# Efficient Resource-Constrained Training of Transformers via Subspace Optimization

**Le-Trung Nguyen**   **Enzo Tartaglione**   **Van-Tam Nguyen**
LTCI, Télécom Paris, Institut Polytechnique de Paris, France
`{name.surname}@telecom-paris.fr`

## Abstract

As AI increasingly shapes daily life, energy consumption and data privacy have become pressing concerns. On-device learning trains models directly on edge devices, cutting energy consumption and safeguarding data privacy. However, the expanding scale of modern neural networks creates a major obstacle for on-device training. Although prior work has concentrated on compact convolutional architectures, we instead apply subspace-based training to transformer models. Motivated by the idea that a model's essential information lies in a fixed subspace, we introduce Weight-Activation Subspace Iteration (WASI), a method that mitigates the memory bottleneck of backpropagation and boosts inference efficiency in transformer models by restricting training to this subspace. Our results demonstrate that WASI maintains accuracy comparable to vanilla training while reducing memory usage by up to $62\times$ and computational cost (FLOPs) by up to $2\times$. On a Raspberry Pi 5, WASI achieves roughly $1.4\times$ faster training and inference than vanilla training. The code is available at https://github.com/Le-TrungNguyen/ICLR2026-WASI.git.

## 1 Introduction

On-device learning has recently emerged as a promising research direction, enabling deep learning models to be fine-tuned directly on resource-constrained edge devices. This approach addresses critical issues such as privacy and energy consumption, improves scalability, and places control of AI capabilities directly "in user's hands" (Dhar et al., 2021). Prior work on on-device learning has largely focused on vision tasks using convolutional neural network models, primarily because of their compact architectures (Lin et al., 2022; Nguyen et al., 2024; Yang et al., 2023b; Quélennec et al., 2024; Bragagnolo et al., 2022; Nguyen et al., 2025).

In many real-world applications, however, transformer-based models have become the de facto choice due to their unique architectural mechanisms (Vaswani et al., 2017). Specifically, these models employ efficient forward propagation through the composition of linear layers, process large-scale data in parallel, and alleviate the vanishing gradient problem thanks to self-attention – key advantages that make them well-suited for handling long-range dependencies, whether in extended text sequences or high-resolution images. Notable examples of

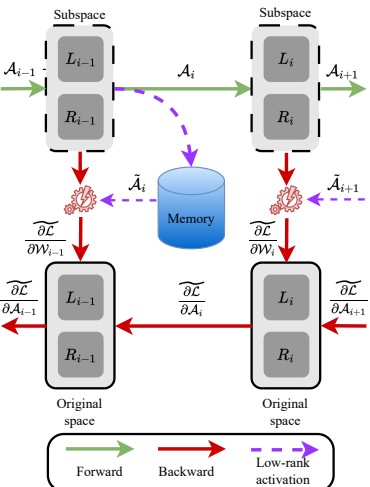

Figure 1: Overview of WASI in a single training iteration.

such models include GPT (Brown et al., 2020), Gemini (Team et al., 2023), LLaMA (Touvron et al., 2023), and DeepSeek (Liu et al., 2024a). Nevertheless, these mechanisms make training and deployment of transformer models resource-intensive. This is even worse when considering the on-device learning context, where models need to be trained on separate edge devices and are often resource-constrained.

A significant fraction of training costs arises from backpropagation, especially the memory and computations needed for storing tensors in model layers (Lin et al., 2022). Various research has emerged to address the inefficiencies of backpropagation and enable learning directly on devices. For instance, Lin et al. (2022) demonstrated the feasibility of fine-tuning a predefined subnetwork under a 256KB memory constraint device while still maintaining competitive performance. Quélennec et al. (2024) took this further by dynamically adapting the subnetwork during training rather than relying on a static one, leading to better accuracy within tight memory budgets. Beyond the scope of on-device learning, many methods aim to reduce training overhead through parameter-efficient approaches, such as LoRA (Hu et al., 2022) and its variants (Xu et al., 2023; Zhang et al., 2023; Hayou et al., 2024; Liu et al., 2024b). While these techniques successfully limit the number of parameters updated at training time, they often overlook the cost of storing intermediate calculations (activation maps). Nguyen et al. (2024) address this by compressing activation maps under a controlled information-loss constraint, but lack robust memory budget control and incur considerable compression overhead.

None of these methods enhances the neural architecture itself, and inference proceeds as usual, resulting in high deployment costs on edge devices. This issue has been further addressed by ASVD (Yuan et al., 2023) and FWSVD (Hsu et al., 2022), which employ truncated Singular Value Decomposition (SVD) to decompose the model architecture, but lack a theoretical basis for choosing which singular values to truncate. Subsequently, SVD-LLM (Wang et al., 2024) was developed to overcome this limitation and outperforms the aforementioned approaches. However, these methods are specifically designed for large language models (LLMs) and are not readily applicable to all vision transformer-based models (see Appendix. A.4). Another similar effort, ESPACE (Sakr & Khailany, 2024), requires access to a downstream dataset, which is not feasible in on-device learning scenarios.

Inspired by prior studies on the stability of parameter subspaces during fine-tuning (Radiya-Dixit & Wang, 2020; Li & Zhang, 2021), we present WASI (Fig. 1), the first method for efficient *model-activation-decomposition-aware training*. WASI enables transformer models to be fine-tuned and executed entirely in a low-rank representation, substantially reducing hardware costs and making vision transformer tasks feasible on edge devices. We assess its effectiveness on vision transformer models, including the Swin Transformer (SwinT) (Liu et al., 2021), the Vision Transformer (ViT) (Dosovitskiy et al., 2020), and even TinyLlama (Zhang et al., 2024).

Our main contributions are summarized as follows.

- Based on the previous studies, we formulate that the essential information of a model parameters resides in a stable subspace throughout fine-tuning (Sec. 3.3), which is then verified in Sec. 4.2.
- Leveraging this hypothesis, we propose Weight-Activation Subspace Iteration (WASI) in Sec. 3.3 to effectively compress the model architecture under a controlled information-loss constraint.
- We showcase the effectiveness of our approach through extensive experiments on multiple tasks (Sec. 4.3 and Sec. 4.4).

## 2 RELATED WORKS

In this section, we review low-rank decomposition techniques as applied to two key components of deep learning models: model weights and activation maps. Other research directions such as compact model design, quantization, sparsification, and knowledge distillation also exist, but they fall outside the scope of this work–*low-rank decomposition* (Cheng et al., 2017; Deng et al., 2020). Therefore they are not discussed here (see Appendix A.5 for details).

**Low-rank Decomposition for Model Weights.** Low-rank approximation methods for model weights have been extensively studied and can generally be categorized into two main approaches: *Low-rank Adapters* and *Low-rank Models*.

LoRA (Hu et al., 2022) is the most prominent example of the first category, which introduces an additional low-rank adapter while freezing the original model architecture. This strategy can reduce the number of trainable parameters by up to four orders of magnitude, but comes with two notable drawbacks. During training, memory usage grows because both the frozen weights and the new

adapter must co-exist in memory. At inference time, the adapter is merged back into the model, resulting in inference performance that is identical to the original model, and thus losing the computational advantages of low-rank decomposition.

Low-rank Models are an alternative line of research that factorizes the weight matrices themselves and trains only the low-rank components, enabling inference to run directly on the compressed representation. Methods such as ASVD (Yuan et al., 2023) and FWSVD (Hsu et al., 2022) achieve this by applying truncated SVD to each layer. These approaches, however, lack a theoretical link between the truncation loss and model performance loss, which is latter addressed by SVD-LLM (Wang et al., 2024). It is important to note that, except for SVD-LLM, all aforementioned methods are specifically tailored for LLMs, and even SVD-LLM cannot be directly applied to all vision transformer-based models with activation maps of four or more dimensions (see Appendix A.4).

**Low-rank Decomposition for Activation Maps.** In addition to model weights, activation maps are a major contributor to memory consumption during training. Gradient Filter (Yang et al., 2023b) is a pioneering work that addresses this issue in on-device learning by generating approximated versions of activation maps through pooling operations with a predefined patch size, aiming to reduce memory usage and FLOPs during fine-tuning. However, this method is limited to convolutional models, and also has the drawback of the accumulated errors as fine-tuning progresses deeper into the model (Nguyen et al., 2024). To overcome this drawback, Nguyen et al. (2024) introduced Activation Map Compression (AMC), which applies High-Order Singular Value Decomposition (HOSVD) to compress activation maps while controlling the information loss via a threshold parameter $\varepsilon$. While AMC achieves impressive memory savings up to $120\times$, it incurs significant computational overhead due to the need for full HOSVD at every iteration. Additionally, the varying ranks required to meet the error threshold lead to fluctuating memory usage, which complicates deployment on devices with fixed memory budgets.

Activation Subspace Iteration (ASI) (Nguyen et al., 2025) addresses both of these issues. Instead of controlling the reconstruction error, ASI fixes the activation ranks using a perplexity-based heuristic. This approach stabilizes memory usage throughout fine-tuning and allows for replacing the expensive HOSVD with subspace iteration. As a result, ASI preserves the high compression ratio of AMC while reducing computational cost by up to $252.65\times$. On a Raspberry Pi 5, fine-tuning with ASI is $1.56\times$ faster than vanilla training when being tested on a highly compact convolutional model.

Beyond this scope, LBP-WHT (Yang et al., 2023b) has also been explored. However, it focuses solely on reducing computational cost during training by applying the Walsh-Hadamard Transformation to tensors in gradient computations, and does not address memory bottlenecks.

Our proposed WASI overcomes the limitations posed by prior works. Hypothesizing the stability of the essential subspace of model weights, we introduce a novel method that simultaneously compresses the model architecture and activation maps while carefully controlling information loss throughout the fine-tuning process. This capability makes it feasible to fine-tune transformer-based models in on-device learning scenarios.

## 3 METHOD

In this section, we first identify the computational bottlenecks of training and inference (Sec. 3.1). Next, we review how activation maps can be efficiently compressed (Sec. 3.2). We then introduce a compression-aware-training strategy for both model weights and activation maps that controls information loss (Sec. 3.3). Finally, we analyze the computational complexity of our method and discuss its practical advantages (Sec. 3.4).

### 3.1 BOTTLENECKS IN TRAINING AND INFERENCE

Consider a deep transformer-based model, where $i$ denotes the index of a linear layer. This layer is represented by a weight matrix $\mathcal{W}_i \in \mathbb{R}^{O_i \times I_i}$, which takes as input a tensor $\mathcal{A}_i \in \mathbb{R}^{B \times N_i \times I_i}$ and produces an output tensor $\mathcal{A}_{i+1} \in \mathbb{R}^{B \times N_i \times O_i}$. Here, $B$ is the batch size, $N_i$ is the sequence length (or number of tokens), $I_i$ is the input feature dimension, and $O_i$ is the output feature dimension. We denote the dimensionality of the input as $\mathcal{D}_i = \{B, N_i, I_i\}$.

During the forward pass (similarly in inference), the output of this layer is computed as:

$$\mathcal{A}_{i+1} = \mathcal{A}_i \mathcal{W}_i^\top, \tag{1}$$

---

**Algorithm 1** Weight Subspace Iteration - WSI at iteration $t$

---

1: **Input:**
    Weight $\mathcal{W}_{i,(t)}$ at iteration $t$,
    Explained variance threshold $\varepsilon \in [0, 1]$.
2: **Function:**
3:  **if** $t = 0$ **then**
4:   $L_{i,(t)}, R_{i,(t)} = \text{SVD}\left(\mathcal{W}_{i,(t)}, \varepsilon\right)$                       (see Eq. 5, Eq. 6, and Eq. 7)
5:  **else**
6:   $R_{i,(t)}^T = \mathcal{W}_{i,(t)}^T \cdot L_{i,(t-1)}$
7:   $L_{i,(t)} = \text{Orthogonalize}\left(\mathcal{W}_{i,(t)} \cdot R_{i,(t)}^T\right)$          (Using Gram-Schmidt)
8:  **endif**
9: **return** $L_{i,(t)}, R_{i,(t)}$

---

where $\top$ denotes the matrix transpose. Eq. 1 presents a batch matrix multiplication applied over the last two dimensions of $\mathcal{A}_i$; that is, for each sample in the batch and each token, a matrix multiplication is performed between a $1 \times I_i$ vector and the transposed weight matrix of size $I_i \times O_i$.

Similarly, in the backward pass the chain rule of backpropagation is computed as follows:

$$\frac{\partial \mathcal{L}}{\partial \mathcal{W}_i} = \frac{\partial \mathcal{L}}{\partial \mathcal{A}_{i+1}}^\top \cdot \frac{\partial \mathcal{A}_{i+1}}{\partial \mathcal{W}_i} = \frac{\partial \mathcal{L}}{\partial \mathcal{A}_{i+1}}^\top \cdot \mathcal{A}_i, \tag{2}$$

$$\frac{\partial \mathcal{L}}{\partial \mathcal{A}_i} = \frac{\partial \mathcal{L}}{\partial \mathcal{A}_{i+1}} \cdot \frac{\partial \mathcal{A}_{i+1}}{\partial \mathcal{A}_i} = \frac{\partial \mathcal{L}}{\partial \mathcal{A}_{i+1}} \cdot \mathcal{W}_i, \tag{3}$$

where $\mathcal{L}$ is the loss computed at the output of the model. Apparently, to compute $\frac{\partial \mathcal{L}}{\partial \mathcal{W}_i}$ and $\frac{\partial \mathcal{L}}{\partial \mathcal{A}_i}$ during the backward pass, $\mathcal{A}_i$ and $\mathcal{W}_i$ must be stored during the forward pass. The large size of these tensors is the primary cause of memory bottlenecks during backpropagation (Lin et al., 2022). Additionally, it also contributes to high inference costs, as multiplying between large $\mathcal{W}_i$ and $\mathcal{A}_i$ requires significant computational resources.

### 3.2 ACTIVATION SUBSPACE ITERATION

Here, we recap how activation maps can be decomposed by subspace iteration. Given an activation memory budget $\mathcal{B}$, ASI performs brute-force optimization before fine-tuning to find an optimal rank vector $\mathbf{r}_i \in \mathbb{N}^3$ for each layer such that the resulting memory does not exceed $\mathcal{B}$. Then, for each mode $m \in \{1, 2, 3\}$, the activation map $\mathcal{A}_i$ is unfolded into a matrix $A_{i,m} \in \mathbb{R}^{a_{i,m} \times b_{i,m}}$, where $(a_{i,m}, b_{i,m}) = \left(\mathcal{D}_{i,m}, \prod_{j \neq m} \mathcal{D}_{i,j}\right)$.

Vogels et al. (2019) showed that warm-started subspace iteration matches SVD performance on stable tensors at much lower cost. Exploiting the stability of activation maps during fine-tuning, ASI applies this technique to each $A_{i,m}$. The resulting approximation takes the form of a Tucker decomposition (Tucker, 1966):

$$\mathcal{A}_i \approx \tilde{\mathcal{S}}_i \times_1 \tilde{U}_i^{(1)} \times_2 \tilde{U}_i^{(2)} \times_3 \tilde{U}_i^{(3)}, \tag{4}$$

where $\tilde{\mathcal{S}}_i \in \mathbb{R}^{\mathbf{r}_{i,1} \times \mathbf{r}_{i,2} \times \mathbf{r}_{i,3}}$ is the core tensor, representing a compressed version of $\mathcal{A}_i$, and each factor matrix $\tilde{U}_i^{(m)} \in \mathbb{R}^{a_{i,m} \times \mathbf{r}_{i,m}}$ contains the principal components along the $m^{\text{th}}$ mode.

Consequently, instead of storing all $\Theta_{\text{space}}\left(\prod_{m=1}^3 \mathcal{D}_{i,m}\right)$ elements of $\mathcal{A}_i$, ASI reduces the storage requirement to $\Theta_{\text{space}}\left(\prod_{m=1}^3 \mathbf{r}_{i,m} + \sum_{m=1}^3 \mathcal{D}_{i,m}\mathbf{r}_{i,m}\right)$.

Details of the algorithm can be found in Appendix A.2.

### 3.3 WEIGHT - ACTIVATION SUBSPACE ITERATION

**Stability of Model Parameters Subspace.** While prior work has shown that over-parameterized models in fact reside in a low-dimensional intrinsic subspace (Aghajanyan et al., 2020; Li et al.,

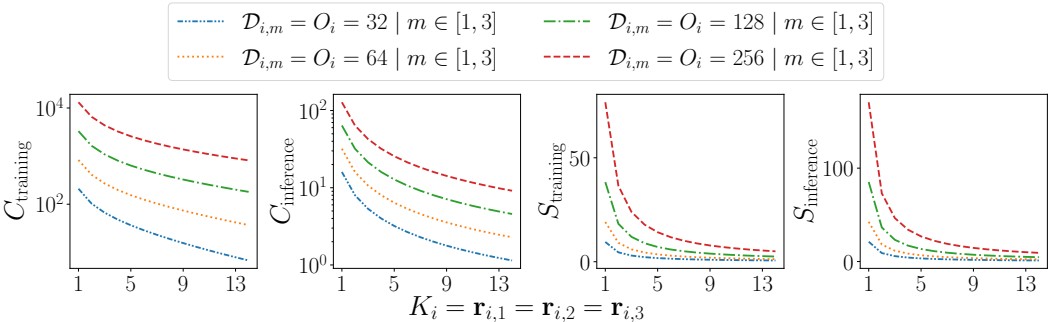

Figure 2: For the linear layer $i$ with a single data batch of size $B$, given varying dimensions of $\mathcal{W}_i$ and $\mathcal{A}_i$ and different values of $\mathbf{r}_{i,m}$, $C_{\text{training}}$ and $C_{\text{inference}}$ illustrate the evolution in compression rates for training and inference, respectively; while $S_{\text{training}}$ and $S_{\text{inference}}$ forecast the speedup ratios for these processes.

2018), we further observe that fine-tuning introduces only minor updates at each training step due to the use of a small learning rate. As a result, our key insight is that the intrinsic subspace remains relatively stable after each training iteration and can therefore be reused in the following one (confirmed in Sec. 4.2 - Fig. 3). This is supported by the findings of Radiya-Dixit & Wang (2020) and Li & Zhang (2021), who showed that the fine-tuned models are close in parameter space to the pre-trained counterpart.

**Weight Subspace Iteration.** Besides activation maps, model parameters (weights) $\mathcal{W}_i$ are another major source of memory bottlenecks during training. To address this, we propose a low-rank weight decomposition strategy that projects each weight tensor into a smaller subspace at every training iteration, thereby preserving the meaningful subspace. The method works as follows:

Step 1. For the weight tensor $\mathcal{W}_i$ at layer $i$, its SVD form is given by:

$$\mathcal{W}_i = U_i \Sigma_i V_i^T, \qquad U_i \in \mathbb{R}^{O_i \times O_i}, \quad \Sigma_i \in \mathbb{R}^{O_i \times I_i}, \quad V_i \in \mathbb{R}^{I_i \times I_i}, \tag{5}$$

where $\Sigma_i$ is a diagonal matrix containing $r_i$ singular values $s_{i,j \in [1,r_i]}$, and $r_i$ is the rank of $\mathcal{W}_i$. As shown in Eq. 3, truncating $U_i$, $\Sigma_i$, and $V_i^T$ inevitably introduces error into $\frac{\partial \mathcal{L}}{\partial \mathcal{A}_i}$, which then propagates backward during training. In other words, low-rank decomposition of the weights affects model convergence due to the accumulation of truncation error.

To control this effect, we constrain the truncation error by enforcing a target explained variance threshold $\varepsilon$, similar to the strategy used in Nguyen et al. (2024). Specifically, we measure the variance explained by the $j^{\text{th}}$ singular value as $\sigma_{i,j}^2 = s_{i,j}^2 / \sum_k s_{i,k}^2$. Assuming the singular values are sorted in descending order ($s_{i,j} \geq s_{i,k}, \forall j \leq k$), the optimal rank is defined as the smallest integer $K_i \in [1, r_i]$ such that $\sum_{j=1}^{K_i} \sigma_{i,j}^2 \geq \varepsilon$. We then identify the essential subspace with rank $K_i$ of $\mathcal{W}_i$, represented by $L_i$ and $R_i$ such that:

$$\mathcal{W}_i \approx \tilde{\mathcal{W}}_i = L_i R_i, \tag{6}$$

where

$$L_i = U_{i,(K_i)} \Sigma_{i,(K_i)}, \quad R_i = V_{i,(K_i)}^T \mid U_{i,(K_i)} \in \mathbb{R}^{O_i \times K_i}, \quad \Sigma_{i,(K_i)} \in \mathbb{R}^{K_i \times K_i}, \quad V_{i,(K_i)} \in \mathbb{R}^{I_i \times K_i}. \tag{7}$$

Step 2. Performing full SVDs at every iteration, however, is computationally prohibitive for on-device training (Nguyen et al., 2025). Leveraging the stability of parameter subspaces established above, $\Sigma_i$ can be expected to remain relatively stable. Thus, for a fixed $\varepsilon$, the optimal rank $K_i$ should also remain consistent (verified in Sec. 4.2). Consequently, instead of recomputing the SVD at every iteration, we compute it once at the beginning to determine the essential subspace. Subspace iteration is applied during training to minimize computational overhead. We refer to this method as **W**eight **S**ubspace **I**teration (WSI), with the full procedure outlined in Algorithm 1.

**Weight-Activation Subspace Iteration.** While WSI reduces weight-related overhead, activation maps also dominate memory usage in backpropagation (Sec. 3.1). Previous work has shown that most of the energy in activation maps is concentrated in the first few principal components across all modes (Nguyen et al., 2024). Such a distribution makes them highly compressible while

still achieving high-fidelity reconstruction (confirmed in Sec. 4.2 - Fig. 4 and Sec. 4.3). Motivated by this property, we propose a unified framework in which both weights and activations are compressed under stable low-rank subspaces. Specifically, we redesign ASI with two improvements: (i) a dynamic-programming strategy that determines $\mathbf{r}_i$ by minimizing memory usage under a target pre-tuning perplexity, rather than relying on a fixed budget $\mathcal{B}$, thereby reducing the search cost from exponential to linear (Appendix A.2); and (ii) an extension to support 3D activation tensors (Appendix A.1).

Together, WSI and ASI form the proposed Weight-Activation Subspace Iteration (WASI), a novel framework for low-rank training that jointly leverages the stability of both weights and activations. Under this scheme, the forward and backward passes are computed as follows:

$$\mathcal{A}_{i+1} = \mathcal{A}_i R_i^T L_i^T, \tag{8}$$

$$\widetilde{\frac{\partial \mathcal{L}}{\partial \mathcal{W}_i}} = f_{\text{LR}} \left( \tilde{\mathcal{A}}_i, \widetilde{\frac{\partial \mathcal{L}}{\partial \mathcal{A}_{i+1}}} \right), \tag{9}$$

$$\widetilde{\frac{\partial \mathcal{L}}{\partial \mathcal{A}_i}} = \widetilde{\frac{\partial \mathcal{L}}{\partial \mathcal{A}_{i+1}}} \cdot L_i R_i, \tag{10}$$

where $f_{\text{LR}}(.)$ denotes a linear operator applied in the low-rank space (see Appendix A.1). With learning rate $\eta$, the weight update is then computed as:

$$L_i R_i = L_i R_i + \eta \cdot \widetilde{\frac{\partial \mathcal{L}}{\partial \mathcal{W}_i}}. \tag{11}$$

### 3.4 Memory Efficiency and Computational Complexity Analysis

For simplicity, we assume that the same optimal rank is applied to both $\mathcal{A}i$ and $\mathcal{W}i$. By varying this value, we can predict total memory usage and speedup for WASI compared to vanilla training (Fig. 2). As model size grows and the optimal rank decreases, WASI delivers greater memory compression ($C_{\text{training}}$, $C_{\text{inference}}$) and speedup ($S_{\text{training}}$, $S_{\text{inference}}$), a property especially valuable in on-device learning where models are typically over-parameterized and reside in low-dimensional subspaces (Aghajanyan et al., 2020; Li et al., 2018). Conversely, as the optimal rank increases, WASI's computational cost approaches that of vanilla training, and the speedup ratios converge to 1, reflecting the upper bound set by vanilla training.

Detailed derives of $C_{\text{training}}$, $C_{\text{inference}}$, $S_{\text{training}}$, and $S_{\text{inference}}$ can be found in Appendix A.3.

## 4 Experiments

In this section, we present experiments designed to demonstrate the effectiveness of WASI. We begin by outlining the experimental setup in Sec. 4.1. Then, in Sec. 4.2, we conduct experiments to validate the assumptions introduced in Sec. 3.3 and Sec. 3.3. Sec. 4.3 compares WASI with various state-of-the-art methods across multiple datasets. Finally, all methods are evaluated in a real-world deployment scenario (Sec. 4.4. All simulation experiments are conducted using PyTorch 1.13.1 on an NVIDIA Quadro RTX A4500 with 20 GB of VRAM, while on-device experiments are run on a Raspberry Pi 5 equipped with a Cortex-A76 CPU and 8 GB of RAM.

### 4.1 Experimental Setup

Our goal is to enable on-device training of transformer models, where networks pretrained on large-scale datasets are fine-tuned locally with task-specific data (Murshed et al., 2021). We evaluate WASI on image classification using ViT and SwinT, both pretrained on ImageNet-1K (Deng et al., 2009), across five downstream datasets: CIFAR-10/100 (Krizhevsky, 2009), CUB (Wah et al., 2011), Flowers (Nilsback & Zisserman, 2008), and Pets (Zhang et al., 2022).

Comparisons are made against three directly comparable baselines at the time of conducting experiments: ASI, SVD-LLM, and vanilla training (as discussed in Secs. 1, 2, Appendix A.5). We measure memory and computation costs during training and inference, focusing on linear layers within multi-perceptron blocks for fair comparison with previous methods (extended results with attention layers in Appendix B.3). All experiments are run with the same set of hyperparameters, detailed in Appendix B.1.

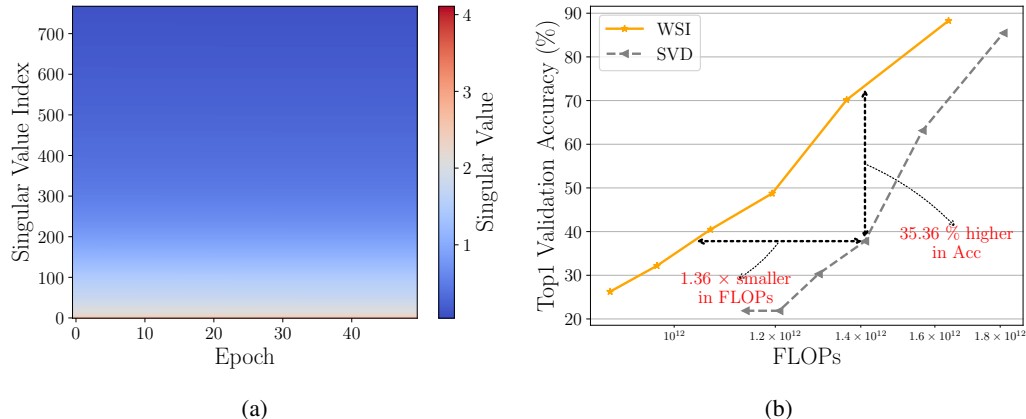

(a)                                                                 (b)

Figure 3: When fine-tuning ViT on the Pets dataset, **(a)** illustrates the evolution of singular values of $\mathcal{W}_6$ across epochs; **(b)** compares WSI and full SVD in terms of accuracy and training FLOPs under varying explained variance thresholds, $\varepsilon \in \{0.4, 0.5, 0.6, 0.7, 0.8, 0.9\}$.

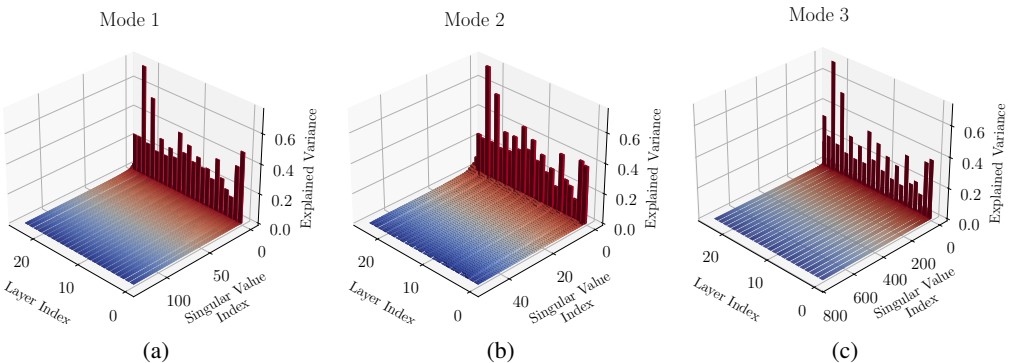

(a)                                        (b)                                        (c)

Figure 4: Explained variance of each singular value of $\mathcal{A}_i$ across all of its modes when fine-tuning ViT on the Pets dataset.

### 4.2 PRELIMINARY RESULTS

In these experiments, we focus on fine-tuning ViT model using Pets dataset.

**Stability of Layer Ranks.** We apply truncated SVD to the weight tensors of the linear layers within ViT's MLP blocks at each training iteration. We constrain the decomposition by setting $\varepsilon = 0.8$ and monitor the layer ranks $K_i$ throughout the course of training. As shown in Fig. 3a, we observe that the ranks exhibit remarkable stability across epochs. This observation validates our insight in Sec. 3.3, confirming the stability of layer ranks during training.

**WSI vs SVD.** Next, we compare two strategies: (1) reapplying truncated SVD at every training iteration, and (2) WSI. We evaluate their performance across a range of $\varepsilon$ values - specifically, 0.4, 0.5, 0.6, 0.7, 0.8, and 0.9 - with each value represented by a different marker in Fig. 3b. The results demonstrate that incorporating subspace iteration through WSI leads to a significant reduction in computational complexity compared to performing a full SVD at every iteration. Specifically, WSI requires $1.36\times$ fewer FLOPs than SVD to achieve the same level of accuracy. Moreover, when both methods are constrained to use the same amount of FLOPs, WSI outperforms SVD by approximately $35\%$ in terms of accuracy. This result verifies that reusing the subspace in subsequent training iterations does not degrade model convergence.

**Explained Variance Distribution of Activation Maps.** Fig. 4 illustrate the explained variances $\sigma_{i,j,m}$ of each singular value $j$ in mode $m$ of the activation map $\mathcal{A}_i$. As anticipated in Sec. 3.3, most activation-map energy lies in the first few singular values, which capture the key information during fine-tuning.

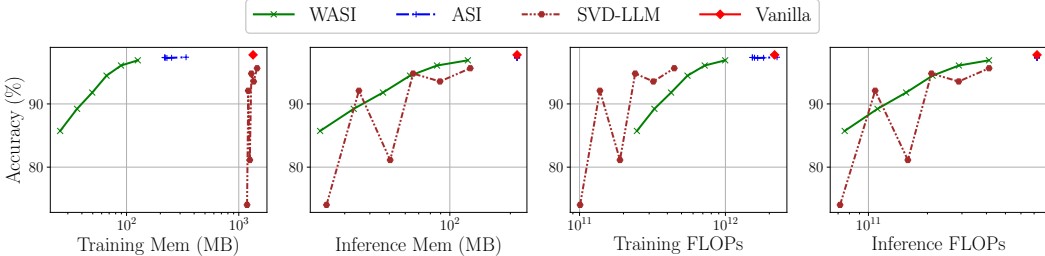

Figure 5: Resource consumption during fine-tuning and inference of ViT on the CIFAR-10 dataset. Each marker in the plots corresponds to a different compression rate, with the red diamond indicating vanilla training.

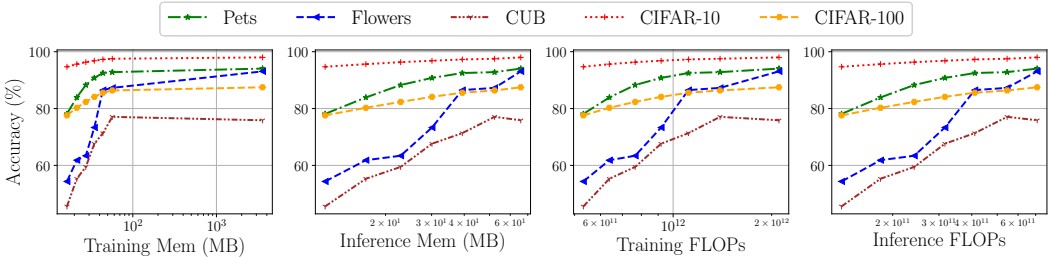

Figure 6: Resource consumption when applying WASI for fine-tuning and inference of SwinT across different datasets. Each marker along the curves represents a different compression rate, while the final marker on each curve corresponds to vanilla training.

## 4.3    MAIN RESULTS

**ViT on CIFAR-10.** Fig. 5 presents the results of fine-tuning a ViT pretrained on ImageNet-1K using CIFAR-10. Each curve for WASI and ASI contains six markers, corresponding to explained variance thresholds $\varepsilon \in \{0.4, 0.5, 0.6, 0.7, 0.8, 0.9\}$ from left to right. The red diamond indicates vanilla training, and for fairness, the same compression ratios are applied to SVD-LLM.

WASI achieves up to $100\times$ higher memory efficiency than SVD-LLM at similar accuracy, owing to its avoidance of LoRA adapters. Its accuracy also improves steadily as $\varepsilon$ increases. In contrast, at the lowest compression rates (last two markers), SVD-LLM consumes even more memory than vanilla training because of the overhead of storing sub-layer activations.

In terms of computation, LoRA adapters allow SVD-LLM to achieve the lowest FLOPs, followed by WASI, which jointly compresses weights and activations into a low-rank subspace. Since ASI only compresses activations while keeping weights intact, its computational cost is higher, and at $\varepsilon = 0.9$, it even exceeds vanilla training (confirmed in Tab. 2). On the other hand, ASI maintains stable accuracy across compression rates, supporting the stability assumption discussed in Sec. 3.3. At inference, both WASI and SVD-LLM achieve similar memory/FLOPs savings, while ASI resembles vanilla since the architecture is unchanged.

**SwinT on Multiple Datasets.** Fig. 6 compares WASI and vanilla across datasets, additional baselines are in Appendix B.3. Each marker along a curve from left to right, indicates different $\varepsilon \in \{0.4, \ldots, 1.0\}$, with $1.0$ as vanilla. Across all datasets, WASI consistently provides a better accuracy-efficiency trade-off. At $\varepsilon = 0.9$, it matches vanilla accuracy while cutting memory by up to $62\times$ and FLOPs by $1.5\times$, and even surpasses vanilla on CUB.

**WASI on TinyLlama.** The initial goal of WASI was to enable training transformer-based models on edge devices, so we focused on ViT and SwinT. To test its generality, we extended our experiments to TinyLlama, a decoder-only transformer model. The downstream dataset used is BoolQ (Clark et al., 2019). Due to limited resources, we only fine-tune up to the last 5 layers of the model and set the WASI $\varepsilon$ to $0.1$. All other training hyperparameters followes the same configuration as in our previous experiments. For comparison, we log the resource consumption only at the layers that are fine-tuned. The results are shown in Fig. 7.

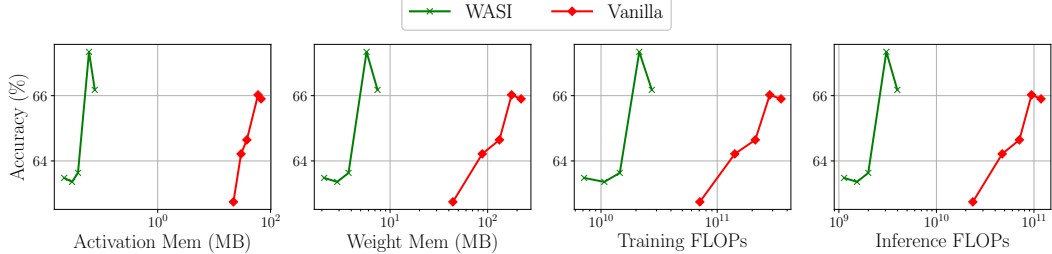

Figure 7: Performance of WASI vs. vanilla training when fine-tuning TinyLlama on BoolQ. Each marker indicates the number of layers fine-tuned from the last layer upward: the marker closest to the y-axis of each figure corresponds to fine-tuning only the last layer, the next marker corresponds to the last two layers, and so on.

WASI again outperforms vanilla: activation and weight memory drop by up to $953.86\times$ and $30.12\times$, while training and inference FLOPs fall by $13.11\times$ and $30.27\times$, all without accuracy loss.

Additional results, including ViT on more datasets and extended baselines for SwinT are in Appendix B.3.

## 4.4 ON-DEVICE LATENCY

We evaluate the practical efficiency of WASI on resource-constrained hardware by fine-tuning ViT on CIFAR-10 using a Raspberry Pi 5. Fig. 8 reports the average time required to complete a single iteration of both training and inference across different explained variance thresholds $\varepsilon \in \{0.4, 0.5, 0.6, 0.7, 0.8, 0.9\}$, along with vanilla training.

As expected, the runtime for both training and inference under WASI increases as $\varepsilon$ becomes larger. This trend aligns with the intuition that higher $\varepsilon$ values retain more information and thus result in higher-rank approximations, which require more compute and memory.

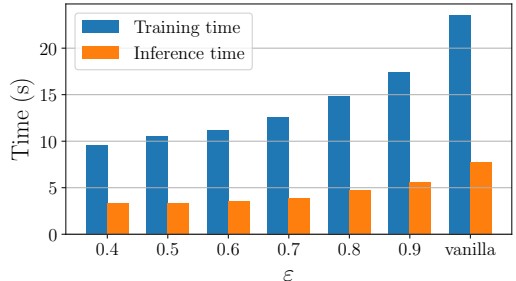

Figure 8: Training and inference time per iteration for ViT on CIFAR-10 (batch size = 128) using a Raspberry Pi 5, measured under different explained variance thresholds $\varepsilon$. The final marker on each curve represents vanilla training.

However, despite this increase, WASI consistently outperforms vanilla training in terms of speed. For instance, even at $\varepsilon = 0.9$, which corresponds to the least aggressive compression setting in this experiment, WASI remains approximately $1.4\times$ faster than vanilla training. Thus, WASI delivers clear benefits even when preserving much of the original information.

Importantly, WASI helps to reduce runtime without causing significant accuracy degradation, as discussed in earlier sections. This ability makes it a strong candidate for the deployment of transformer-based model in real-world on-device learning scenarios, where computational resources are severely constrained. Further numerical results can be found in Appendix. B.3.

## 5 CONCLUSION

In this work, we introduced WASI, an efficient training method for resource-constrained fine-tuning of transformer models. Assuming that essential parameter information lies in a stable low-dimensional subspace, WASI applies SVD and subspace iteration to obtain low-rank approximations of both weights and activations during each training iteration. This yields significant gains in memory and computation while tightly controlling information loss.

Building on prior theory and validated through extensive experiments, WASI outperforms state-of-the-art methods, reducing training memory usage by up to $62\times$ and achieving $1.4\times$ speedup over vanilla training on a Raspberry Pi 5. These results show the potential of WASI for enabling on-device learning with transformers, a domain traditionally dominated by CNNs. While our experiments focus on transformers, the underlying principles apply broadly to any neural network trained with backpropagation.

ACKNOWLEDGEMENT

Part of this work was funded by Hi!PARIS Center on Data Analytics and Artificial Intelligence, by the European Union's Horizon Europe Research and Innovation Programme under grant agreement No. 101120237 (ELIAS - European Lighthouse of AI for Sustainability) and No. 101120657 (ENFIELD - European Lighthouse to Manifest Trustworthy and Green AI), by the French National Research Agency (ANR) in the framework of the IA Cluster project "Hi! PARIS Cluster 2030" (ANR-23-IACL-005), the NF-NAI project (ANR-22-PEFT-0003) and NF-FITNESS project (ANR-22-PEFT-0007) as part of France 2030.

REPRODUCIBILITY STATEMENT

Detailed description of our algorithm is provided in Sec. 3.3, Appendix A.1, and Appendix A.2. Full details of the training policy, including hyperparameters, datasets, and other configurations, are presented in Appendix B.1. Code to reproduce the main experiments is included in the Supplementary Material zip file. We commit to open-sourcing the complete code upon acceptance of this paper.

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

# A ADDITIONAL THEORETICAL DETAILS

## A.1 DETAILS OF BACKPROPAGATION IN LOW-RANK SUBSPACE

In this section, we explain the definition of $f_{\text{LR}}(.)$ in Eq. 9. For simplicity, we denote the activation tensor $\mathcal{A}_i$ as $\mathcal{I}$, the output gradient $\frac{\partial \mathcal{L}}{\partial \mathcal{W}_i}$ as $\Delta \mathcal{W}$, and the gradient with respect to the output $\frac{\partial \mathcal{L}}{\partial \mathcal{A}_{i+1}}$ as $\Delta \mathcal{Y}$.

**3D Activation Maps.** For each activation map $\mathcal{I} \in \mathbb{R}^{B \times N \times I}$, applying ASI with optimal rank $\mathbf{r} \in \mathbb{R}^3$, resulting in the following approximation:

$$\tilde{\mathcal{I}}_{b,n,i} = \sum_{r_1=1}^{\mathbf{r}_1} \sum_{r_2=1}^{\mathbf{r}_2} \sum_{r_3=1}^{\mathbf{r}_3} \tilde{S}_{r_1,r_2,r_3} \tilde{U}_{b,r_1}^{(1)} \tilde{U}_{n,r_2}^{(2)} \tilde{U}_{i,r_3}^{(3)} \tag{12}$$

which transforms the weight gradient calculation (Eq. 9) into:

$$\widetilde{\Delta \mathcal{W}_{o,i}} = \sum_{b=1}^{B} \sum_{n=1}^{N} \sum_{o=1}^{O} \sum_{i=1}^{I} \tilde{\mathcal{I}}_{b,n,i} \widetilde{\Delta \mathcal{Y}}_{b,n,o} \tag{13}$$

$$= \sum_{b=1}^{B} \sum_{n=1}^{N} \sum_{o=1}^{O} \sum_{i=1}^{I} \sum_{r_1=1}^{\mathbf{r}_1} \sum_{r_2=1}^{\mathbf{r}_2} \sum_{r_3=1}^{\mathbf{r}_3} \tilde{S}_{r_1,r_2,r_3} \tilde{U}_{b,r_1}^{(1)} \tilde{U}_{n,r_2}^{(2)} \tilde{U}_{i,r_3}^{(3)} \widetilde{\Delta \mathcal{Y}}_{b,n,o} \tag{14}$$

By reordering and grouping terms, we obtain:

$$\mathcal{Z}_{n,o,r_1}^{(1)} = \sum_{b=1}^{B} \widetilde{\Delta \mathcal{Y}}_{b,n,o} \tilde{U}_{b,r_1}^{(1)}, \tag{15}$$

$$\mathcal{Z}_{r_1,r_3,n}^{(2)} = \sum_{r_2=1}^{\mathbf{r}_2} \tilde{S}_{r_1,r_2,r_3} \tilde{U}_{n,r_2}^{(2)}, \tag{16}$$

$$\mathcal{Z}_{r_1,i,n}^{(3)} = \sum_{r_3=1}^{\mathbf{r}_3} \mathcal{Z}_{r_1,r_3,n}^{(2)} \tilde{U}_{i,r_3}^{(3)}, \tag{17}$$

$$\widetilde{\Delta \mathcal{W}}_{o,i} = \sum_{n=1}^{N} \sum_{r_1=1}^{\mathbf{r}_1} \mathcal{Z}_{n,o,r_1}^{(1)} \mathcal{Z}_{r_1,i,n}^{(3)}. \tag{18}$$

**4D Activation Maps.** In some transformer-based models, such as SwinT, the activation maps are 4D: $\mathcal{I} \in \mathbb{R}^{B \times H \times W \times I}$. When applying ASI with optimal rank $\mathbf{r} \in \mathbb{R}^4$, the activation map is approximated as:

$$\tilde{\mathcal{I}}_{b,h,w,i} = \sum_{r_1=1}^{\mathbf{r}_1} \sum_{r_2=1}^{\mathbf{r}_2} \sum_{r_3=1}^{\mathbf{r}_3} \sum_{r_4=1}^{\mathbf{r}_4} \tilde{S}_{r_1,r_2,r_3,r_4} \tilde{U}_{b,r_1}^{(1)} \tilde{U}_{h,r_2}^{(2)} \tilde{U}_{w,r_3}^{(3)} \tilde{U}_{i,r_4}^{(4)} \tag{19}$$

Reorganizing the terms yields:

$$\widetilde{\Delta \mathcal{W}_{o,i}} = \sum_{b=1}^{B} \sum_{h=1}^{H} \sum_{w=1}^{W} \sum_{o=1}^{O} \sum_{i=1}^{I} \tilde{\mathcal{I}}_{b,h,w,i} \widetilde{\Delta \mathcal{Y}}_{b,h,w,o} \tag{20}$$

$$= \sum_{b=1}^{B} \sum_{h=1}^{H} \sum_{w=1}^{W} \sum_{o=1}^{O} \sum_{i=1}^{I} \sum_{r_1=1}^{\mathbf{r}_1} \sum_{r_2=1}^{\mathbf{r}_2} \sum_{r_3=1}^{\mathbf{r}_3} \sum_{r_4=1}^{\mathbf{r}_4} \tilde{S}_{r_1,r_2,r_3,r_4} \tilde{U}_{b,r_1}^{(1)} \tilde{U}_{h,r_2}^{(2)} \tilde{U}_{w,r_3}^{(3)} \tilde{U}_{i,r_4}^{(4)} \widetilde{\Delta \mathcal{Y}}_{b,h,w,o}$$

$$\tag{21}$$

Again, by reordering and grouping operator, Eq. 21 becomes:

$$\mathcal{Z}^{(1)}_{r_1,h,w,o} = \sum_{b=1}^{B} \widetilde{\Delta\mathcal{Y}}_{b,h,w,o} \tilde{U}^{(1)}_{b,r_1}, \tag{22}$$

$$\mathcal{Z}^{(2)}_{r_1,h,r_3,r_4} = \sum_{r_2=1}^{\mathbf{r}_2} \tilde{S}_{r_1,r_2,r_3,r_4} \tilde{U}^{(2)}_{h,r_2}, \tag{23}$$

$$\mathcal{Z}^{(3)}_{r_1,h,r_3,o} = \sum_{r_3=1}^{\mathbf{r}_3} \mathcal{Z}^{(1)}_{b,h,w,o} \tilde{U}^{(3)}_{w,r_3}, \tag{24}$$

$$\mathcal{Z}^{(4)}_{r_1,h,i,r_3} = \sum \mathcal{Z}^{(2)}_{r_1,h,r_3,r_4} \tilde{U}^{(4)}_{i,r_4}, \tag{25}$$

$$\widetilde{\Delta\mathcal{W}}_{o,i} = \sum_{h=1}^{H} \sum_{r_1=1}^{\mathbf{r}_1} \sum_{r_3=1}^{\mathbf{r}_3} \mathcal{Z}^{(3)}_{r_1,h,r_3,o} \mathcal{Z}^{(4)}_{r_1,h,i,r_3}. \tag{26}$$

## A.2 ADDITIONAL ALGORITHMIC DETAILS

**The $i$-Mode Product Operation.** The $i$-mode product "$\times_i$" of a $n^{th}$-order tensor $\mathcal{G} \in \mathbb{R}^{P_1 \times P_2 \times \cdots \times P_n}$ and a matrix $B \in \mathbb{R}^{Q \times P_i}$ is a $n^{th}$-order tensor $\mathcal{R} \in \mathbb{R}^{P_1 \times \cdots \times P_{i-1} \times Q \times P_{i+1} \times \cdots \times P_n}$ can be expressed as:

$$\mathcal{R}_{p_1,\ldots,p_{i-1},q,p_{i+1},\ldots,p_n} = \mathcal{G} \times_i Q = \sum_{p_i=1}^{P_i} g_{p_1,p_2,\ldots,p_n} b_{q,p_i}. \tag{27}$$

**Subspace Iteration.** Here, we present how can activation maps be compressed by the subspace iteration technique of PowerSGD (Vogels et al., 2019). The underlying idea remains the same: a single step of subspace iteration (Stewart & Miller, 1975) is used to obtain a fast low-rank approximation of a matrix, which in this case corresponds to an unfolding of the activation maps along their respective modes. As noted in the original work, however, a single iteration often yields an approximation that is not sufficiently accurate. To address this, Vogels *et al.* propose initializing each step with the low-rank approximation obtained in the previous iteration.

This reuse is particularly well justified in activation maps. While activation maps evolve as the network parameters are updated, the changes across consecutive iterations remain small. This stability arises from the Lipschitz continuity of activation functions (Virmaux & Scaman, 2018) together with the incremental nature of parameter updates during optimization. By reusing the previous approximation, the sequence of activation maps across iterations can be effectively smoothed and reduced the variance of the low-rank approximation compared to the case without reuse. A formal proof of this property is provided in (Vogels et al., 2019).

**Perplexity for Activation Compression.** The perplexity of activation $\mathcal{P}_{\mathcal{A}_i}$ is defined as the difference between $\frac{\partial\mathcal{L}}{\partial\mathcal{W}_i}$ and $\widetilde{\frac{\partial\mathcal{L}}{\partial\mathcal{W}_i}}$ to represent the error introduced when performing activation compression.

Estimating perplexity over compression levels requires HOSVD on four-dimensional activation tensors, which naively induces an exponential number of rank combinations across modes and layers. To avoid this combinatorial blow-up, the search over ranks is replaced by a set $\mathcal{E}$ of explained-variance thresholds. Let $\mathcal{E} \in (0,1]^E$ denote the $E$ thresholds evaluated; each threshold yields a consistent set of truncation ranks for all considered layers. The procedure has two steps:

Step 1. For each threshold $\varepsilon_j \in \mathcal{E}$, run a forward pass on a held-out batch. At layer $i$, cache the original activation $(\mathcal{A}_i)_j$ and its low-rank approximation $(\tilde{\mathcal{A}}_i)_j$, obtained via HOSVD with components truncated according to $\varepsilon_j$.

Step 2. During backpropagation, compute the exact gradient $(\frac{\partial\mathcal{L}}{\partial\mathcal{W}_i})_j$ and its approximated counterpart $(\widetilde{\frac{\partial\mathcal{L}}{\partial\mathcal{W}_i}})_j$. The layer-wise perplexity at threshold $\varepsilon_j$ is the Frobenius norm of their difference:

$$\mathcal{P}_{i,j} = \left\| \left(\frac{\partial\mathcal{L}}{\partial\mathcal{W}_i}\right)_j - \left(\widetilde{\frac{\partial\mathcal{L}}{\partial\mathcal{W}_i}}\right)_j \right\|_F. \tag{28}$$

---

**Algorithm 2** ASI for layer $i$ with set of rank $\mathbf{r}_i \in \mathcal{R}_{\text{opt}}$

---

1: **Input:**
   Activation map $\mathcal{A}_i^{(t)} \in \mathbb{R}^{B \times N_i \times O_i}$ at epoch $t$.
   Target ranks for 3 modes $\mathbf{r}_i \in \mathbb{N}^3 \cap [1, \min(a_{i,m}, b_{i,m})]$, where $(a_{i,m}, b_{i,m})$ is the shape of $\mathcal{A}_i^{(t)}$ at mode $m$.
2: **Function:**
3:   Initialize $S_i = \mathcal{A}_i^{(t)}$
4: **for** $m = 1$ to 3 **do**
5:   $A_{i,m} = \text{unfold } \mathcal{A}_i^{(t)}$ along mode $m$, $A_{i,m} \in \mathbb{R}^{a_{i,m} \times b_{i,m}}$
6:   **if** $t = 0$ **then**
7:     Initialize $V \in \mathbb{R}^{b_{i,m} \times \mathbf{r}_{i,m}}$ from an i.i.d. standard normal distribution.
8:   **else**
9:     $V = A_{i,m}^T U_{i,m}^{(t)}$
10:  **end if**
11:  $U_{i,m}^{(t)} = \text{Orthogonalize}(A_{i,m} V)$
12:  $S_i = S_i \times_m U_{i,m}^{(t)}$
13: **end for**
14: **return** $S_i, U_{i,m}^{(t)}$ with $m = 1, \dots, 3$

---

This process is repeated across all layers, resulting in a perplexity matrix $\mathcal{P} \in \mathbb{R}^{N \times E}$ and a corresponding rank tensor $\mathcal{R}^{N \times E \times 3}$, containing the selected ranks across the 3 modes of activation tensors for each combination of layer and explained variance threshold.

**Rank Selection.** Given a memory budget $\mathcal{B}$ for activations over the fine-tuned layer set $\mathcal{F}$, the goal is to choose mode-wise truncation ranks that satisfy the budget while minimizing total perplexity. A recursive backtracking routine identifies an index set $\mathcal{J} \in \mathbb{N} \cap [1, E]$ and the corresponding optimal ranks $\mathbf{r}$ such that

$$\mathbf{r}_{i,m} = \mathcal{R}_{i,j,m} \mid j \in \mathcal{J}^*, \tag{29}$$

$$\mathcal{J}^* = \arg \min_{\mathcal{J}, \sum_{i=1}^{|\mathcal{F}|} M_i \leq \mathcal{B}} \left( \sum_{i=1}^{|\mathcal{F}|} \sum_{j \in \mathcal{J}} \mathcal{P}_{i,j} \right) \tag{30}$$

where

$$M_i = \prod_{m=1}^{3} \mathbf{r}_{i,m} + \sum_{m=1}^{3} \mathcal{D}_{i,m} \mathbf{r}_{i,m} \tag{31}$$

denotes the activation memory induced by the chosen ranks $\mathbf{r}_i$ for layer $i$.
Detail of ASI is shown Algorithm 2.

**Rank Selection in WASI.** For WASI, $\mathcal{J}^*$ is found such that:

$$\mathcal{J}^* = \arg \min_{\mathcal{J}} \left[ \sum_{i=1}^{|\mathcal{F}|} \left( \prod_{m=1}^{3} \mathcal{R}_{i,j,m} + \sum_{m=1}^{3} \mathcal{D}_{i,m} \mathcal{R}_{i,j,m} \right) \right] \tag{32}$$

## A.3 DETAILS OF COMPUTATIONAL SPEEDUP AND SPACE COMPLEXITY

**Computational Speedup.** We derive the computational speedup as the ratio between the total FLOPs required for vanilla training and those required by WASI.

First, the number of FLOPs required to perform forward pass in vanilla training (Eq. 1) is:

$$F_{\text{vanilla}} \approx 2BN_i I_i O_i \tag{33}$$

Meanwhile, the backward pass includes Eq. 2 and Eq. 3, costing:

$$B_{\text{vanilla}} \approx 4BN_i I_i O_i \tag{34}$$

In contrast, WASI performs the forward pass in a low-rank subspace (Eq. 8) with a complexity of:

$$F_{\text{WASI}} \approx 2BN_iK_i(I_i + O_i) \tag{35}$$

However, this does not account for the overhead from weight subspace decomposition (Algorithm 1) and ASI decomposition. These add the following costs:

$$O_{\text{WSI}} = 4I_iO_iK_i + 2O_iK_i^2, \tag{36}$$

$$O_{\text{ASI}} = \sum_{m=1}^{3} \left( 4dd'\mathbf{r}_{i,m} + 2d\mathbf{r}_{i,m}^2 \right), \quad \text{where } d = \mathcal{D}_{i,m}, ; d' = \mathcal{D}_i \setminus \{d\} \tag{37}$$

The backward pass in WASI follows Eq. 10, Eq. 15, Eq. 16, Eq. 17, and Eq. 18, with a total FLOPs cost of:

$$B_{\text{WASI}} = \underbrace{2BN_iK_i(I_i + O_i)}_{\text{Eq. 10}} + \underbrace{BN_iO_i\mathbf{r}_{i,1} + \mathbf{r}_{i,1}\mathbf{r}_{i,2}\mathbf{r}_{i,3}N_i + \mathbf{r}_{i,1}\mathbf{r}_{i,3}I_iN_i + \mathbf{r}_{i,1}I_iO_iN_i}_{\text{Eq. 15 to Eq. 18}} \tag{38}$$

The speedup ratios between vanilla training and WASI are defined as:

$$S_{\text{training}} = \frac{F_{\text{vanilla}} + B_{\text{vanilla}}}{F_{\text{WASI}} + O_{\text{WSI}} + O_{\text{ASI}} + B_{\text{WASI}}} \tag{39}$$

$$S_{\text{inference}} = \frac{F_{\text{vanilla}}}{F_{\text{WASI}}} \tag{40}$$

**Memory Usage.** In vanilla training, the total memory consists of the memory for weights and the memory for storing activations:

$$M_{\text{vanilla}}^{(\mathcal{W}_i)} = I_iO_i \tag{41}$$

$$M_{\text{vanilla}}^{(\mathcal{A}_i)} = BN_iI_i \tag{42}$$

In WASI, these become:

$$M_{\text{WASI}}^{(\mathcal{W}_i)} = K_i(I_i + O_i) \tag{43}$$

$$M_{\text{WASI}}^{(\mathcal{A}_i)} = \prod_{m=1}^{3} \mathbf{r}_{i,m} + \sum_{m=1}^{3} \mathcal{D}_{i,m}\mathbf{r}_{i,m} \tag{44}$$

Thus, the memory reduction ratios between vanilla and WASI are:

$$C_{\text{training}} = \frac{M_{\text{vanilla}}^{(\mathcal{W}_i)} + M_{\text{vanilla}}^{(\mathcal{A}_i)}}{M_{\text{WASI}}^{(\mathcal{W}_i)} + M_{\text{WASI}}^{(\mathcal{A}_i)}} \tag{45}$$

$$C_{\text{inference}} = \frac{M_{\text{vanilla}}^{(\mathcal{W}_i)}}{M_{\text{WASI}}^{(\mathcal{W}_i)}} \tag{46}$$

Similar ratios can be derived for the case of 4D activation maps.

### A.4 LIMITATIONS OF SVD-LLM

We analyze the 3D activation map $\mathcal{A}_i \in \mathbb{R}^{B \times N_i \times I_i}$, as considered throughout the paper. The core idea of SVD-LLM is to incorporate "Truncation-aware Data Whitening" mechanism that enables a direct relationship between singular values and the resulting compression loss.

To do this, SVD-LLM whitens the activation using a transformation of the form $S_i^{-1}X_i$, where $X_i \in \mathbb{R}^{N_i \times I_i}$ is obtained by summing $\mathcal{A}_i$ over the batch dimension. The goal is to make the transformed activation orthonormal, i.e., $(S_i^{-1}X_i)(S_i^{-1}X_i)^T = I$. Here, $S_i$ is computed via Cholesky decomposition on $X$.

SVD is then applied to the transformed weight matrix $\mathcal{W}_i S_i$, resulting in $U_i$, $\Sigma_i$, and $V_i$. Based on the optimal rank $K_i$ found by a desired compression ratio, the smallest singular values in $\Sigma_i$ are truncated (denoted by $\Sigma_{i,(K_i)}$) to obtain two low-ranking matrices:

$$\mathcal{W}'^{(u)}_i = U_{i,(K_i)}\Sigma^{1/2}_{i,(K_i)}, \quad \mathcal{W}'^{(v)}_i = \Sigma^{1/2}_{i,(K_i)}V^T_{i,(K_i)}S^{-1}_i \tag{47}$$

and the final compressed matrix is given by:

$$\tilde{\mathcal{W}}_i = \mathcal{W}'^{(u)}_i \mathcal{W}'^{(v)}_i = U_{i,(K_i)}\Sigma_{i,(K_i)}V^T_{i,(K_i)}S^{-1}_i. \tag{48}$$

However, a current limitation of SVD-LLM is that "Truncation-aware Data Whitening" is only defined for 3D activation maps. It does not generalize to 4D activations, which are common in certain architectures such as SwinT. As a result, SVD-LLM cannot be directly applied to models that rely on 4D activation structures.

## A.5 OTHER RESEARCH DIRECTIONS

Model compression and acceleration have become central topics in deep learning research, with efforts ranging from hardware improvements to algorithmic techniques. On the algorithmic side, the main directions include *compact model design*, *quantization*, *sparsification*, *knowledge distillation*, and *low-rank decomposition* (Cheng et al., 2017; Deng et al., 2020). The shared goal of these approaches is to shrink the size of neural networks while keeping their accuracy intact. Among them, low-rank decomposition has gained particular attention because it combines solid theoretical foundations with practical advantages for deployment (Xinwei et al., 2023). Originally developed in systems theory and signal processing (Markovsky, 2008), low-rank methods were first used in deep learning to compress fully connected layers through singular value decomposition (SVD) (Xue et al., 2013). Since then, more advanced forms such as generalized Kronecker product decomposition (GKPD) (Hameed et al., 2022), semi-tensor products (STP) (Zhao et al., 2021), and applications to vision transformers (Yang et al., 2023a) have pushed the field forward. Progress is largely measured in terms of compression ratio, inference speed, and power efficiency, with the challenge being to achieve these gains while minimizing performance loss.

Note that the research directions mentioned above are orthogonal to each other and can be combined to leverage their respective strengths.

## B ADDITIONAL EXPERIMENTAL DETAILS

### B.1 DETAILS OF EXPERIMENTAL SETUP

To ensure a fair comparison, we follow the same experimental setup as described in (Nguyen et al., 2024). The key details are as follows:

**General hyperparameters.** All models are first pretrained on ImageNet-1K, then fine-tuned on a different downstream dataset using an $80\% - 20\%$ train-validation split in $50$ epochs. We use cross-entropy loss and optimize the models with SGD. The initial learning rate is set to $0.05$ and decayed using a cosine annealing schedule. Momentum is set to $0$, and weight decay is fixed at $1 \times 10^{-4}$. We apply L2 gradient clipping with a threshold of $2.0$. For data augmentation, we use random resizing, horizontal flipping, normalization, and a mini-batch size of $128$.

**ASI.** We follow the same strategy as proposed in (Nguyen et al., 2025). Specifically, in our main experiments, we apply AMC with a range of $\varepsilon$ values: $0.4$, $0.5$, $0.6$, $0.7$, $0.8$, and $0.9$. For each value of $\varepsilon$, we record the peak activation memory consumption of AMC and use it as the activation memory budget for ASI. The perplexity of ASI is also measured based on these $\varepsilon$ values.

**SVD-LLM.** Unlike WASI, SVD-LLM compresses models based on a fixed compression ratio. To compare fairly, we compute the compression ratios achieved by WASI at each $\varepsilon$ and use the same ratios for SVD-LLM. We also adopt the same LoRA adapter settings as used in their original paper: $\alpha = 16$ and rank $= 8$.

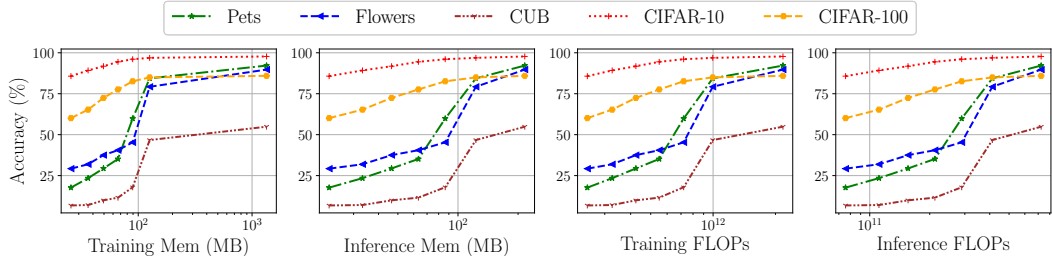

Figure 10: WASI performance when fine-tuning ViT across multiple datasets. In each plot, markers from left to right represent increasing values of $\varepsilon$; the rightmost marker corresponds to vanilla training.

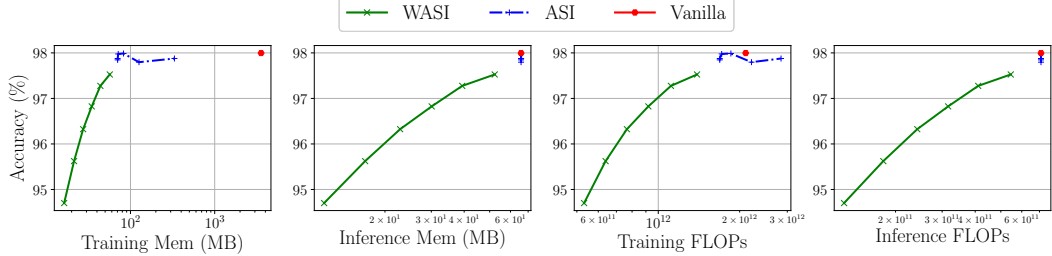

Figure 11: Comparison of different methods when fine-tuning SwinT on CIFAR-10. For each plot, markers from left to right correspond to increasing values of $\varepsilon$.

## B.2 VARIANCE ACROSS DIFFERENT RANDOM SEEDS

Fig. 9 presents the results of fine-tuning a ViT model pretrained on ImageNet-1K using WASI on Pets dataset. We test different values of $\varepsilon$ from $\{0.4, 0.5, 0.6, 0.7, 0.8, 0.9\}$, with each marker on the plot (from left to right) corresponding to one of these settings. For each $\varepsilon$, we report the average accuracy and peak training memory usage over three random seeds (233, 234, and 235), with error bars representing standard deviation.

As expected, increasing $\varepsilon$ leads to higher accuracy and greater memory usage, reflecting the trade-off between compression and performance. Importantly, the variance across seeds is minimal, which is reasonable given that WASI mainly uses deterministic components such as SVD, Gram-Schmidt orthogonalization, and matrix multiplications. Therefore, we fix the random seed to 233 in all other experiments.

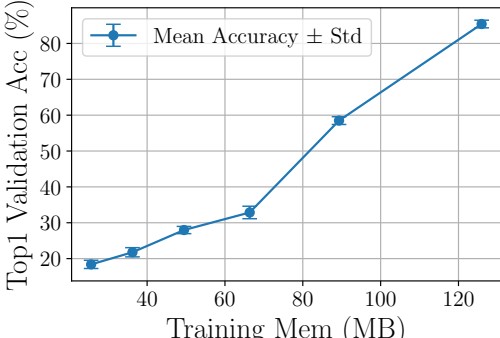

Figure 9: WASI performance on the with different $\varepsilon$ values, showing mean accuracy and memory usage across three random seeds. Error bars represent standard deviation.

## B.3 ADDITIONAL RESULTS

**Additional Transformer-based Results.** We conduct experiments similar to those in Sec. 4.3. Fig. 10 and Fig. 11 show consistent trends with our earlier findings. Note that due to the architectural design of SwinT, which generates 4-dimensional activation maps, the "Truncation-Aware Data Whitening" mechanism used in SVD-LLM is not applicable (see Appendix. A.4). Therefore, this method is excluded from the experiments in Fig. 11.

The accuracy of WASI improves steadily as $\varepsilon$ increases, demonstrating the effectiveness of controlling compression error through the explained variance threshold. Overall, WASI achieves up to one

Table 1: Performance of WASI with different $\varepsilon$ values on all linear layers (including attention blocks and MLP blocks) of ViT using the CIFAR-10 dataset. Note that $\varepsilon = 1.0$ corresponds to vanilla training.

| $\varepsilon$ | Train Mem (MB) | Infer Mem (MB) | Train FLOPs | Infer FLOPs | Acc. (%) |
|---|---|---|---|---|---|
| 0.4 | 39.39 | 32.43 | $3.92 \times 10^{11}$ | $1.08 \times 10^{11}$ | 68.99 |
| 0.5 | 54.02 | 47.06 | $4.99 \times 10^{11}$ | $1.57 \times 10^{11}$ | 78.53 |
| 0.6 | 72.28 | 65.32 | $6.33 \times 10^{11}$ | $2.19 \times 10^{11}$ | 85.50 |
| 0.7 | 95.70 | 88.74 | $8.06 \times 10^{11}$ | $2.97 \times 10^{11}$ | 90.52 |
| 0.8 | 127.85 | 120.89 | $1.04 \times 10^{12}$ | $4.05 \times 10^{11}$ | 94.07 |
| 0.9 | 179.61 | 172.65 | $1.43 \times 10^{12}$ | $5.79 \times 10^{11}$ | 96.24 |
| 1.0 | 2349.00 | 324.00 | $3.26 \times 10^{12}$ | $1.09 \times 10^{12}$ | 97.32 |

order of magnitude reduction in training memory when fine-tuning ViT, with similarly favorable results observed in terms of computational cost and inference memory.

**WSI on Convolutional Neural Network.** In this experiment, we extend the application scope of WSI to convolutional neural networks. Specifically, we use MCUNet (Lin et al., 2022), pretrained on ImageNet-1K, with Pets dataset as the downstream task. WSI is applied to fine-tune the last 1 to 4 convolutional layers of the model. When $\varepsilon = 0.9$, applying WSI unexpectedly increases the weight memory. This is due to the fact that the optimal rank found is too high, resulting in principal components whose total number of elements exceeds that of the original weight tensor. In contrast, for smaller values of $\varepsilon$ (0.75 and 0.8), applying WSI to more layers leads to reduced memory usage for the weights, as expected, at the cost of some accuracy degradation. However, this trade-off is not worthwhile, as the memory savings are marginal and the convolutional layers are already highly compact by design.

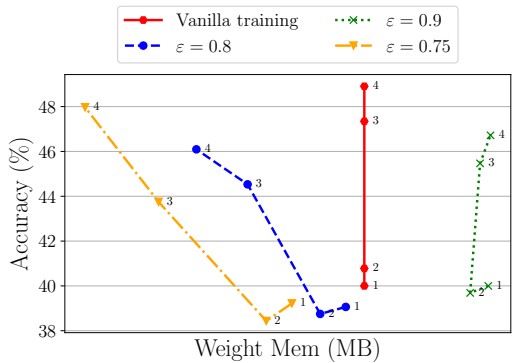

Figure 12: Performance of WSI when applied to fine-tune MCUNet (pretrained on ImageNet-1K) on Pets dataset. The number next to each marker indicates how many convolutional layers WSI was applied to.

**Extending WASI to multi-head attention projections.** For a fair comparison with state-of-the-art baselines, we relied on their efficient implementations and applied WASI only to the linear layers in MLP blocks for the main experiments. To demonstrate the broader applicability of WASI, we then extended the experiments to attention blocks as well. The setup was identical to that described in Fig. 5, and the results are summarized in Tab. 1.

**Numerical On-device Latency.** Tab. 2 reports the latency results of WASI, ASI, and vanilla training when fine-tuning ViT on a Raspberry Pi 5. We used the same setup as in Sec. 4.4. As the compression rank increases (i.e., larger $\varepsilon$), ASI becomes progressively slower, eventually even slower than vanilla training. In contrast, WASI consistently maintains its speed. These findings are further confirmed by Fig. 5 and Fig. 10.

**Latency on Multiple Edge Devices.** We conducted additional experiments using the same setup described in Sec. 4.4. The results are shown in Tab. 3.

**On-device Energy Consumption Results.** We reran the experiment in Sec. 4.4 by fine-tuning a ViT on a single minibatch of 128 CIFAR-10 samples initialized with ImageNet-pretrained weights. Energy consumption was measured on a Jetson Orin using its on-board INA3221 power sensor. Tab. 4 reports the energy consumption for one inference pass and one training iteration.

Table 2: Comparison of inference and training time (s) when applying WASI, ASI, and vanilla training to fine-tune ViT on Raspberry Pi 5 at different $\varepsilon$ values.

| $\varepsilon$ | WASI | | ASI | | Vanilla | |
|---|---|---|---|---|---|---|
| | Infer. | Train. | Infer. | Train. | Infer. | Train. |
| 0.4 | 3.15 | 8.49 | 7.69 | 18.35 | – | – |
| 0.5 | 3.35 | 9.54 | 7.76 | 18.38 | – | – |
| 0.6 | 3.56 | 10.21 | 7.94 | 19.22 | – | – |
| 0.7 | 3.88 | 11.61 | 7.85 | 21.30 | – | – |
| 0.8 | 4.49 | 13.75 | 7.71 | 22.57 | – | – |
| 0.9 | 5.58 | 16.57 | 7.91 | 25.52 | – | – |
| 1.0 | – | – | – | – | 7.87 | 23.87 |

Table 3: On-device latency of fine-tuning ViT on one minibatch of 128 CIFAR-10 samples initialized with ImageNet-pretrained weights. We report the time for one inference pass and one training iteration on three edge devices.

| $\varepsilon$ | Jetson Orin | | Jetson Nano | | Raspberry Pi 4 | |
|---|---|---|---|---|---|---|
| | Infer (s) | Train (s) | Infer (s) | Train (s) | Infer (s) | Train (s) |
| 0.4 | 1.60 | 5.73 | 5.94 | 71.30 | 5.01 | 16.32 |
| 0.5 | 1.97 | 6.79 | 9.30 | 91.73 | 5.55 | 18.03 |
| 0.6 | 2.26 | 7.61 | 12.84 | 85.43 | 6.57 | 20.65 |
| 0.7 | 2.80 | 8.88 | 19.22 | 93.94 | 7.95 | 24.59 |
| 0.8 | 3.56 | 10.68 | 20.36 | 117.91 | 9.78 | 29.26 |
| 0.9 | 4.57 | 13.58 | 22.67 | 118.60 | 13.14 | 38.15 |
| 1.0 | 6.84 | 21.79 | 29.47 | 241.90 | 20.82 | 65.42 |

## C    LIMITATIONS

Our goal is to enable the training of transformer models on edge devices. So far, our experiments have primarily focused on vision tasks, which allows for direct comparison with existing work in this domain. While our experiments with TinyLlama demonstrate the potential of WASI for LLMs (Fig. 7), current hardware limitations prevent us from evaluating larger-scale models. In future work, we plan to extend our approach to a broader range of tasks, with a particular emphasis on LLMs.

## D    LLM USAGE

We used LLM for grammar editing. All research ideas and the article structure were conceived and developed by the authors.

Table 4: Energy consumption of WASI on Jetson Orin with different $\varepsilon$. Note that $\varepsilon = 1.0$ corresponds to vanilla training.

| $\varepsilon$ | Inference Energy (J) | Training Energy (J) |
|---|---|---|
| 0.4 | 27.86 | 92.42 |
| 0.5 | 29.52 | 96.36 |
| 0.6 | 30.92 | 97.78 |
| 0.7 | 33.67 | 104.21 |
| 0.8 | 38.00 | 110.13 |
| 0.9 | 43.43 | 120.98 |
| 1.0 | 47.51 | 141.87 |

