# OpenReview forum: "Efficient Resource-Constrained Training of Transformers via Subspace Optimization"
_ICLR.cc/2026/Conference — ICLR 2026 Oral_

### Official Review · Reviewer_cFHJ · 2025-10-30

**Soundness:** 3
**Presentation:** 3
**Contribution:** 3
**Rating:** 6
**Confidence:** 3

**Summary:**

This paper proposes Weight–Activation Subspace Iteration (WASI), a method that performs model training entirely within a low-rank subspace of both weights and activation. Experimental results demonstrate that WASI significantly reduces memory usage and roughly halves the computational cost (FLOPs), making on-device training feasible even on constrained hardware such as the Raspberry Pi.

**Strengths:**

- The paper is clearly written and easy to follow.

- The proposed method can be applied to various architectures, including ViT, Swin Transformer, and TinyLlama.

- Experiments convincingly show that WASI drastically reduces memory consumption and FLOPs while maintaining comparable accuracy to full fine-tuning.

**Weaknesses:**

- The core ideas, low-rank approximation of parameters and activations via subspace iteration, have been explored in prior work. Thus, the novelty is somewhat limited, and the main contribution is largely engineering integration rather than conceptual innovation.

- The ablation study indicates that the method can be sensitive to certain hyperparameters and datasets, which may affect stability.

- Experiments are conducted on relatively small datasets, which could lead to higher variance and limit generalization.

- Minor: The figures could be improved, for example, by labeling the ε thresholds more clearly to make the trends easier to interpret.

**Questions:**

- Could the authors evaluate the proposed method on a larger-scale dataset, such as training from scratch on ImageNet-1K? Even if direct on-device testing is infeasible in this scenario, a simulated experiment on full GPUs would provide stronger evidence of the method’s scalability and the effectiveness of the low-rank approximation at large scale.
- Is there a principled way to choose the ε  other than grid sweeping? For example, can it be adaptively determined or estimated from training dynamics?

---

> ### Author Response · Authors · 2025-11-18
> **Author's Response to Reviewer cFHJ - Part I: All Weaknesses**
>
> Dear Reviewer cFHJ,
>
> We appreciate your time and effort in reviewing our paper. Our response exceeds the 5,000-character limit, so we provide it in two separate parts.
> ___
> **[Weakness 1: Novelty of the method.]**
>
> ***"The core ideas, low-rank approximation of parameters and activations via subspace iteration, have been explored in prior work."***
>
> To the best of our knowledge, **no prior work** has proposed jointly applying subspace iteration to both weights and activations during training in the way presented in WASI. If the reviewer is aware of a specific example, we would greatly appreciate a citation, as we could not find any existing work.
>
> ***"the main contribution is largely engineering integration rather than conceptual innovation."***
>
> We thank the reviewer for raising this point, and we would like to clarify our view of the conceptual aspects of the work. Our contributions include:
> - First, we identify and empirically validate the presence of a stable low-dimensional subspace that persists throughout fine-tuning. This observation, shown in Figure 3b and supported by prior analyses (Lines 214 to 236), serves as the foundation for our method.
> - Second, motivated by this insight, we propose a subspace-iteration procedure that decomposes model architecture during each training iteration. To our knowledge, this formulation has not been previously explored.
>
> We hope this clarification helps contextualize the conceptual elements of WASI.
> ___
> **[Weakness 2: Stability of the method.]**
>
> We would appreciate clarification regarding this concern. In our experiments (Figures 5 to 11), WASI exhibits consistent behavior across all evaluated datasets. In particular, the performance-resource trade-off is stable and predictable (as described in Sec. 3.4): smaller $\varepsilon$ yields stronger compression with lower cost and lower accuracy, and larger $\varepsilon$ yields the opposite.
>
> Across all benchmarks, we did not observe instability or dataset-dependent sensitivity. If the reviewer has a specific figure or experimental setting in mind where instability appears, we would be grateful for the pointer so we can examine it more closely.
>
> **[Weakness 3: Experiments are conducted on relatively small datasets.]**
>
> For clarification, on-device learning typically involves using a model that has been pretrained on a large-scale dataset, which is then deployed to an edge device for a specific application. The model is subsequently updated over time through fine-tuning with local, task-specific data [1]. Therefore, using a large-scale dataset for fine-tuning model on edge device is not very representative of this scenario.
>
> We already stated this detail in Lines 314 to 319, and we will make this point more explicit and clearer in the revision.
> ___
> **[Minor: The figures could be improved, for example, by labeling the $\varepsilon$ thresholds more clearly to make the trends easier to interpret.]**
>
> Thank you for this suggestion.
> We agree that clearer labeling would improve readability. What do you think if we annotate each marker with its corresponding $\varepsilon$ value (similar to Figure 12 in our paper)?
> ___
> [1] Murshed, MG Sarwar, et al. "Machine learning at the network edge: A survey." ACM Computing Surveys (CSUR) 54.8 (2021): 1-37.

---

> > ### Comment · Reviewer_cFHJ · 2025-11-27
> >
> > The response addresses my concerns. The key novelty is the emphasis on on-device training, which is a specialized and relevant problem setting. I am therefore raising my score to 8. Also annotating each marker sounds good.

---

> ### Author Response · Authors · 2025-11-18
> **Author's Response to Reviewer cFHJ - Part II: All Questions**
>
> **[Question 1: Training-from-scratch with WASI on ImageNet-1K]**
>
> WASI is specifically designed for **fine-tuning**, similar to other approaches in this direction such as LoRA, SVD-LLM, etc.
>
> Training from scratch with WASI is **not expected** to provide the same benefits as fine-tuning in term of saving resource consumption, for the following reasons:
>
> - Low-rank training is effective only when a meaningful subspace has already been formed (i.e., when using a pretrained model) [2].
> - When training from scratch, the weight tensors need to maintain high rank to preserve enough degrees of freedom for the optimizer to explore the loss landscape.
>
> Therefore, for the same $\varepsilon$, WASI can **compress much more during fine-tuning** than during training from scratch. (*)
> ___
> **Training-from-scratch with ImageNet.**
> Nonetheless, we appreciate the reviewer's request and conducted additional ImageNet-1K training-from-scratch experiments following the same setup as our fine-tuning experiments.
>
> The training curves can be viewed at the following anonymized link: [https://imgur.com/a/3JBhFK9](https://imgur.com/a/3JBhFK9)
>
> The convergence curves demonstrate that even in this setting, WASI's behavior remains consistent: **larger $\varepsilon$ (more information retained) leads to better convergence**.
>
> ___
> **Fine-tuning Vs. Training-from-scratch.**
> To further illustrate point (*), we ran additional ablations on a smaller dataset (Flowers102), comparing: fine-tuning a ViT pretrained on ImageNet-1K vs. training-from-scratch on Flowers102.
>
> In the table below, suffix "1" = fine-tuning, "2" = training-from-scratch.
>
>
> |**$\varepsilon$**|**Weight. Mem. 1 (MB)**|**Acc. 1 (%)**|**Weight. Mem. 2 (MB)**|**Acc. 2 (%)**|
> |---|---|---|---|---|
> |**0.4**|22.63|29.24|62.58|27.34|
> |**0.5**|33.25|31.92|82.88|27.90|
> |**0.6**|46.49|37.50|105.83|29.24|
> |**0.7**|63.37|40.51|132.58|29.24|
> |**0.8**|86.32|45.31|164.68|29.91|
> |**0.9**|122.96|79.24|205.66|30.02|
> |**Vanilla**|216.00|89.73|216.00|29.80|
>
> These results clearly show that, for the same $\varepsilon$, the weight memory in the fine-tuning setting is compressed much more than in the training-from-scratch setting.
>
> Extending WASI to training-from-scratch (a.k.a pretraining) is an interesting direction and we leave this as future work.
>
> ___
> **[Question 2: Strategy to choose $\varepsilon$.]**
>
> Yes. In practice, $\varepsilon$ can be selected based on the resource constraints of the target device.
>
> $\varepsilon$ controls the amount of information preserved after decomposition, and it directly determines both accuracy and resource consumption.
> Because the subspace remains stable during fine-tuning, each $\varepsilon$ consistently leads to a fixed and predictable resource consumption. This allows us to estimate memory usage or latency by running only one iteration of training or inference (not necessarily on the device), which is fast.
>
> Therefore, instead of grid sweeping the entire training process, $\varepsilon$ can be chosen efficiently by testing a few $\varepsilon$ values on a single iteration and selecting the one that satisfies the device’s constraints.
>
> Please refer to **Auto-tuning strategies targeting memory or latency constraints** in **Question 3 of Reviewer qswV** for the detail of how can we choose this $\varepsilon$.
> ___
> [2] Tartaglione, E. (2022, October). The rise of the lottery heroes: why zero-shot pruning is hard. In 2022 IEEE International Conference on Image Processing (ICIP) (pp. 2361-2365). IEEE.

---

### Official Review · Reviewer_qswV · 2025-11-01

**Soundness:** 3
**Presentation:** 3
**Contribution:** 3
**Rating:** 6
**Confidence:** 3

**Summary:**

The paper proposes WASI, a weight–activation subspace iteration framework that trains and runs transformers entirely in a learned low-rank subspace. WASI couples (i) Weight Subspace Iteration (WSI)—initial SVD with explained-variance threshold ε and iterative subspace updates—with (ii) a redesigned Activation Subspace Iteration (ASI) (dynamic-programming rank selection, 3D/4D activations).

**Strengths:**

+ Unlike LoRA-style PEFT that re-inflates at inference, WASI keeps both weights and activations in low rank throughout forward/backward, with explicit FLOPs/memory formulas and subspace-space equations, yielding predictable savings end-to-end.
+ The paper motivates stable layer ranks during fine-tuning  and shows singular-value/rank stability, enabling subspace iteration instead of per-step SVD; empirically WSI dominates repeated SVD in FLOPs/accuracy.
+ The on-device results seems interesting.

**Weaknesses:**

- The edge device used in this paper is only Rasperry pi 5. More edge devices like Jeston Nano should be included and evaluated.

- The final accuracy should be highlighted and reported with Tables. Now it is hard to find it in the draft.

- Main comparisons focus on MLP/linear blocks for fairness with baselines; attention-layer coverage is deferred to appendix. Non-IID, partial-participation, and straggler/client-heterogeneity studies (key for on-device contexts) are not central, leaving external validity to real deployments somewhat open.

**Questions:**

- Can you provide a micro-benchmark of one WASI iteration (basis update, orthogonalization, matmuls) on Pi-5 vs. GPU to attribute the speedups precisely?

- Can you adopt more concrete edge devices for expriments?

- How sensitive are results to ε and the number of subspace-iteration steps? Any auto-tuning strategy that targets a latency or memory cap directly?

---

> ### Author Response · Authors · 2025-11-18
> **Author's Response to Reviewer qswV - Part I: All Weaknesses and Question 2.**
>
> Dear Reviewer qswV,
>
> Thank you for taking the time to review our work. Our response exceeds the 5,000-character limit, so we provide it in three separate parts.
> ___
> **[Weakness 1 and Question 2: More experiments on more edge devices (including Jetson Nano).]**
>
> We conducted additional experiments following the same setup described in Section 4.4 of our paper:
>
> **Raspberry Pi 4**:
> |**$\varepsilon$**|**Inference time (s)**|**Training time (s)**|
> |---|---|---|
> |**0.4**|5.01|16.32|
> |**0.5**|5.55|18.03|
> |**0.6**|6.57|20.65|
> |**0.7**|7.95|24.59|
> |**0.8**|9.78|29.26|
> |**0.9**|13.14|38.15|
> |**Vanilla**|20.82|65.42|
>
> **Jetson Nano**:
> |**$\varepsilon$**|**Inference time (s)**|**Training time (s)**|
> |---|---|---|
> |**0.4**|5.94|71.30|
> |**0.5**|9.30|91.73|
> |**0.6**|12.84|85.43|
> |**0.7**|19.22|93.94|
> |**0.8**|20.36|117.91|
> |**0.9**|22.67|118.60|
> |**Vanilla**|29.47|241.90|
>
> **Jetson Orin**:
> |**$\varepsilon$**|**Inference time (s)**|**Training time (s)**|
> |---|---|---|
> |**0.4**|1.60|5.73|
> |**0.5**|1.97|6.79|
> |**0.6**|2.26|7.61|
> |**0.7**|2.80|8.88|
> |**0.8**|3.56|10.68|
> |**0.9**|4.57|13.58|
> |**Vanilla**|6.84|21.79|
>
> Across devices, WASI consistently reduces training and inference latency compared to vanilla fine-tuning.
>
> These experiments can be reproduced by cloning the `on_device_latency` folder from our supplementary materials on any target device (that supports python/torch) and running the code as described in the included `README.md`.
>
> We will add these results to the revised version of the paper.
> ___
> **[Weakness 2: No table reports on final accuracy.]**
>
> **Why not report final accuracy?**
> We follow the standard evaluation protocol in this field [1,2,3,4,5,etc.] and, for fairness of comparison with prior works, we report only top-1 validation accuracy instead of the final one. This is the traditional metric used in all baseline papers.
>
> **Why not present the results in tables?**
> Because it requires multiple experiments to clearly show the **performance trend** of WASI under different resource budgets. We chose to use Pareto-curve style figures to convey the trade-off more intuitively and avoid cluttering the paper with large tables of numbers.
>
> **Final accuracy with table.**
> As requested, we conducted further experiments over 5 different random seeds (as described in the only **Question of Reviewer ZTZ2**). We report the results in mean ± standard deviation format below. This table summarizes the final validation/training accuracy and final training loss of WASI under different $\varepsilon$ values and vanilla training as a reference baseline:
> |**$\varepsilon$**|**0.4**|**0.5**|**0.6**|**0.7**|**0.8**|**0.9**|**Vanilla**|
> |---|:---:|:---:|:---:|:---:|:---:|:---:|:---:|
> |**Top1 Val Acc (%)**|19.44 ± 1.10|21.76 ± 1.04|27.15 ± 1.01|33.26 ± 0.84|59.96 ± 1.03|85.50 ± 0.68|90.06 ± 0.65|
> |**Final Val Acc (%)**|19.03 ± 1.35|19.96 ± 0.96|25.86 ± 0.83|30.41 ± 0.72|56.31 ± 1.08|84.06 ± 0.33|88.78 ± 0.62|
> |**Final Training Acc (%)**|81.77 ± 1.52|87.62 ± 0.66|96.25 ± 0.60|99.81 ± 0.12|100.00 ± 0.00|100.00 ± 0.00|100.00 ± 0.00|
> |**Final Training Loss**|1.21 ± 0.05|0.99 ± 0.03|0.61 ± 0.04|0.26 ± 0.03|0.02 ± 0.00|0.01 ± 0.00|0.00 ± 0.00|
> |**Training Mem (MB)**|28.60 ± 0.00|36.24 ± 0.00|49.48 ± 0.00|66.36 ± 0.00|89.31 ± 0.00|125.95 ± 0.00|1341.00 ± 0.00|
> |**Inference Mem (MB)**|25.62 ± 0.00|33.25 ± 0.00|46.49 ± 0.00|63.37 ± 0.00|86.32 ± 0.00|122.96 ± 0.00|216.00 ± 0.00|
> |**Training FLOPs**|2.47E+11 ± 0.00|3.26E+11 ± 0.00|4.24E+11 ± 0.00|5.50E+11 ± 0.00|7.21E+11 ± 0.00|9.96E+11 ± 0.00|2.17E+12 ± 0.00|
> |**Inference FLOPs**|1.25E+11 ± 0.00|1.68E+11 ± 0.00|2.22E+11 ± 0.00|2.91E+11 ± 0.00|3.85E+11 ± 0.00|5.38E+11 ± 0.00|7.24E+11 ± 0.00|
> ___
> **[Weakness 3: Attention-layer coverage is deferred to appendix. Concerns about applications for federated learning.]**
>
> ***"Main comparisons focus on MLP/linear blocks for fairness with baselines; attention-layer coverage is deferred to appendix."***
>
> This design choice was intentional and stated explicitly in lines 321 to 323 in our paper:
>
> *"We measure memory and computation costs during training and inference, focusing on linear layers within multi-perceptron blocks for fair comparison with previous methods (extended results with attention layers in Appendix B.3)."*
>
> ***"Non-IID, partial-participation, and straggler/client-heterogeneity studies (key for on-device contexts) are not central, leaving external validity to real deployments somewhat open."***
>
> Our work targets the **single-device fine-tuning** setting for a **specific downstream task**, as stated in lines 314 to 316 in our paper. This setting naturally arises when a single edge device must continuously adapt to its own data stream in a privacy-sensitive environment. Extending WASI to federated learning is an interesting future direction but beyond the scope of our single-device focus. We leave this for future work.

---

> ### Author Response · Authors · 2025-11-18
> **Author's Response to Reviewer qswV - Part II: Questions 1 and 3.**
>
> **[Question 1: Micro-benchmark of one WASI iteration (basis update, orthogonalization, matmuls) on Pi-5 vs. GPU.]**
>
> As stated in Sec. 3.3 and Algorithm 1 of our paper, one WASI iteration consists of:
> - Forward pass: output computation (Eq. 8), matrix multiplications and orthogonalization from WSI (Algorithm 1) and ASI (Algorithm 2).
> - Backward pass (Eqs. 9 and 10).
>
> Below, we provide micro-benchmarks on a Raspberry Pi 5 (CPU) and an NVIDIA P100 GPU.
> The experimental setup follows Sec. 4.4 (ViT fine-tuning on CIFAR-10).
> ___
> **Raspberry Pi 5:**
> |**$\varepsilon$**|**Inference time (s)**|**Training time (s)**|**Forward time (s)**|**Output calculation time (s)**|**Matmuls time (s)**|**Orthogonalization time (s)**|**Backward time (s)**|
> |---|---|---|---|---|---|---|---|
> |**0.4**|3.31|9.54|5.04|3.51|1.46|0.07|4.50|
> |**0.5**|3.38|10.53|5.32|3.68|1.53|0.11|5.22|
> |**0.6**|3.51|11.22|5.74|3.94|1.63|0.17|5.48|
> |**0.7**|3.88|12.62|6.31|4.32|1.73|0.26|6.31|
> |**0.8**|4.72|14.86|7.24|4.94|1.88|0.42|7.62|
> |**0.9**|5.58|17.38|8.63|5.76|2.08|0.79|8.75|
> |**Vanilla**|7.77|23.57|7.89|0.00|0.00|0.00|15.69|
> ___
> **NVIDIA P100:**
> |**$\varepsilon$**|**Inference time (s)**|**Training time (s)**|**Forward time (s)**|**Output calculation time (s)**|**Matmuls time (s)**|**Orthogonalization time (s)**|**Backward time (s)**|
> |---|---|---|---|---|---|---|---|
> |**0.4**|0.00128|0.21087|0.20137|0.00972|0.01938|0.17227|0.00950|
> |**0.5**|0.00125|0.14391|0.13426|0.00389|0.01943|0.11094|0.00965|
> |**0.6**|0.00124|0.13751|0.12738|0.00397|0.01923|0.10418|0.01013|
> |**0.7**|0.00127|0.13636|0.12595|0.00383|0.01977|0.10234|0.01041|
> |**0.8**|0.00280|0.16539|0.15581|0.00394|0.01972|0.13215|0.00958|
> |**0.9**|0.00471|0.15197|0.14169|0.00419|0.02044|0.11706|0.01028|
> |**Vanilla**|0.00073|0.00837|0.00359|0.00000|0.00000|0.00000|0.00477|
> ___
> On **Raspberry Pi 5 (CPU-only)**, WASI is consistently faster than vanilla training. The main speedup comes from performing backward in a low-rank subspace, while the extra orthogonalization and matmuls overhead remains relatively small.
>
> On the **NVIDIA P100 GPU**, our current WASI implementation is slower than vanilla. This is mainly because we still rely on standard PyTorch kernels (see `on_device_latency/custom_op/linear/linear_WASI.py`), which are not fully optimized for GPU execution.
>
> Please note that we never claim GPU speedup in the paper. However, with a dedicated low-level implementation (e.g., custom CUDA/C++ kernels), GPU acceleration is theoretically achievable. Such an optimized implementation requires substantial engineering effort and is beyond the scope of this paper. We leave this for future work.
> ___
> **[Question 3: How sensitive are results to $\varepsilon$ and the number of subspace-iteration steps? Any auto-tuning strategy that targets a latency or memory cap directly?]**
>
> **Sensitivity to $\varepsilon$.**
> Model accuracy and training resource consumption **increase proportionally** with $\varepsilon$. In our method, $\varepsilon$ acts as a hyperparameter that controls how much information is preserved after the low-rank decomposition. A larger $\varepsilon$ keeps more information, leading to higher accuracy but also higher training cost as demonstrated in Figures 5 to 11.
>
> **Sensitivity to the number of subspace-iteration steps.**
> Performing SVD at every iteration is equivalent to recomputing the subspace each iteration. We compared WSI (which performs a single subspace computation) against SVD in Figure 3b (Section 4.2). The results show **no significant accuracy differences** for the same $\varepsilon$, but computing the subspace only once (WSI) **substantially reduces computation cost** ($\approx 1.36\times$ reduction in this experiment).
>
> **Auto-tuning strategies targeting memory or latency constraints.**
> For a fixed $\varepsilon$, WASI yields stable memory usage and computational cost throughout training (due to the stability of the subspace). This allows for a simple engineering strategy to directly target a memory cap:
>
> 1. On the server side, fine-tune the target model with different $\varepsilon$ values. For each $\varepsilon$, run only **one training iteration** to measure resource consumption (In the supplementary code, enabling `log_activation_mem = True` in two files: `trainer_cls.py` for convolutional models and `trainer_cls_linear.py` for transformer models).
> 2. By doing this, we can produce a lookup table mapping $\varepsilon$ to resource consumption.
> 3. Deploy the model to the edge device using the $\varepsilon$ value that satisfies the memory constraint.
>
> A similar strategy can be applied to latency by using the measured FLOPs.

---

> ### Author Response · Authors · 2025-11-18
> **Author's Response to Reviewer qswV - Part III: References.**
>
> [1] Yang, Y., Li, G., & Marculescu, R. (2023). Efficient on-device training via gradient filtering. In Proceedings of the IEEE/CVF Conference on Computer Vision and Pattern Recognition (pp. 3811-3820).
>
> [2] Quélennec, A., Tartaglione, E., Mozharovskyi, P., & Nguyen, V. T. (2024). Towards on-device learning on the edge: Ways to select neurons to update under a budget constraint. In Proceedings of the IEEE/CVF Winter Conference on Applications of Computer Vision (pp. 685-694).
>
> [3] Lin, J., Zhu, L., Chen, W. M., Wang, W. C., Gan, C., & Han, S. (2022). On-device training under 256kb memory. Advances in Neural Information Processing Systems, 35, 22941-22954.
>
> [4] Nguyen, L. T., Quélennec, A., Nguyen, V. T., & Tartaglione, E. (2025). Beyond Low-rank Decomposition: A Shortcut Approach for Efficient On-Device Learning. Forty-Second International Conference on Machine Learning.
>
> [5] Wang, X., Zheng, Y., Wan, Z., & Zhang, M. (2024). Svd-llm: Truncation-aware singular value decomposition for large language model compression. International Conference on Learning Representations.

---

> > ### Comment · Reviewer_qswV · 2025-11-20
> >
> > I highly appreciate the additional experiments added by the author and the efforts the authors making on the rebuttal. As such, I  have increased my final score.

---

### Official Review · Reviewer_ZTZ2 · 2025-11-03

**Soundness:** 3
**Presentation:** 3
**Contribution:** 2
**Rating:** 6
**Confidence:** 4

**Summary:**

WASI is a novel method that applies subspace-based training to Transformer models, primarily aiming for highly efficient on-device fine-tuning by restricting parameter updates to a low-rank subspace that captures the model's essential information. The method overcomes the memory bottleneck of backpropagation and decreases inference latency in transformer models important for edge devices. The results show that WASI maintains accuracy comparable to vanilla training while reducing memory usage by up to 62× and computational cost (FLOPs) by up to 2×. On a Raspberry Pi 5, WASI achieves roughly 1.5× faster training and inference compared to vanilla training.

**Strengths:**

WASI solves an important problem, which is the memory bottleneck of backpropagation. Previous parameter-efficient methods often ignored the activation memory footprint, which scales linearly with batch size and context length. This comes from the fact that WASI iteratively maintains a subspace that efficiently handles both the weights and the intermediate activations.

The method has been shown to reduce memory usage by up to 62 times compared to vanilla training. This is crucial for enabling on-device fine-tuning of large models with severely limited RAM.

By operating within a low-rank subspace, the overall number of floating-point operations (FLOPs) required for both forward and backward passes is significantly reduced. Research reports up to a 2x reduction in computational cost, leading to demonstrably faster training and inference speedups on constrained hardware (e.g., 1.5x faster on devices like the Raspberry Pi).

Unlike methods that aggressively prune models, WASI aims to preserve the model's learning capacity by training within a space believed to contain the essential information. The key strength is maintaining task accuracy comparable to vanilla training, which means the huge efficiency gains do not come at a major performance cost.

I find the writing is clear and the paper is well structured.

The experiment on Raspberry PI is good.

The choice of baseline models, such as VIT, SwinT, on a couple of datasets is satisfactory.

**Weaknesses:**

The method is novel yet incremental. The fundamental idea of restricting model updates to a low-dimensional subspace is established. We see this in earlier methods like Stochastic Weight Averaging (SWA) and subsequent research focused on finding paths or subspaces of high-accuracy models.

The major gains in efficiency are obtained when compared to vanilla and ASI methods. The gains compared to SVD-LLM are minimal, i.e, only for memory consumption during training.

No power consumption results on the device.

The results are shown to work for seed 233. It is mentioned that the variance between the results for different seeds is not high, but there are no results showing that.

**Questions:**

What was the variance between the results with diffence seeds? What is average of results if you try 5 different seeds.

---

> ### Author Response · Authors · 2025-11-18
> **Author's Response to Reviewer ZTZ2**
>
> Dear Reviewer ZTZ2,
>
> Thank you for your thoughtful and constructive comments on our work. Below, we provide our responses to the weaknesses you raised and to your question.
> ___
> **[Weakness 1: Novelty of the method.]**
>
> It is true that *"the fundamental idea of restricting model updates to a low-dimensional subspace is established"*, we cited these works and discuss their limitations (lines 61 to 76).
>
> Other approaches (*) *"focused on finding paths or subspaces of high-accuracy models"* (SWA and its subsequent research as you mentioned), but these focus on improving convergence or generalization rather than reducing training resource consumption.
>
> To the best of our knowledge, **no prior work** leverages the insight that this low-dimensional subspace remains stable during training to design an efficient technique for reducing training costs.
>
> Beside, combining WASI with methods in category (*) could further improve performance. WASI removes high-rank components from both weights and activations, while those methods restrict updates to a low-rank region, preventing important information from shifting into high-rank components.
> ___
> **[Weakness 2: The major gains are for training memory.]**
>
> Memory consumption during training is our main target:
> - Our initial goal is to develop a technique that reduces training memory without introducing computational overhead.
> - As stated in the title of our paper, we introduce WASI primarily as a method for **optimizing the training process**, with a particular focus on on-device training.
>
> Therefore, the efficiency gains we target are specifically during training, especially when comparing with SVD-LLM (the SOTA on weight compression at the time of conducting these experiments).
> ___
> **[Weakness 3: No power consumption results on the device.]**
>
> Thank you for highlighting this.
>
> We reran the experiment in Sec. 4.4: fine-tuning a ViT on one minibatch of 128 CIFAR-10 samples (ImageNet-pretrained). This time, we used a **Jetson Orin with an on-board INA3221 power sensor**.
>
> (We used Jetson Orin instead of Raspberry Pi 5 because the Pi 5 does not expose any on-board power sensor, and at that time we did not have access to an external power meter.)
>
> The results for one inference and one training iteration are shown below:
> |**$\varepsilon$**|**Inference Energy (J)**|**Training Energy (J)**|
> |---|---|---|
> |**0.4**|27.85763751|92.42375369|
> |**0.5**|29.52428364|96.36388178|
> |**0.6**|30.91545642|97.78264146|
> |**0.7**|33.67461275|104.2130695|
> |**0.8**|38.00022106|110.1320345|
> |**0.9**|43.43285632|120.9786178|
> |**Vanilla**|47.51351287|141.8666789|
> These results will be included in the revised version of the paper. We commit to releasing the code for this experiment upon acceptance.
> ___
> **[Weakness 4: No results showing variance across seeds.]**
>
> We used 233 as the main seed for fair comparison with prior work. Results for seeds 233, 234, and 235 are provided in Appendix B.2 (Figure 9).
> ___
> **[Question: Numerical results over 5 seeds.]**
>
> We understand that you want numerical results instead of figure.
>
> Here, we follow the same setup as in Figure 9 - Appendix B.2 in our paper: fine-tuning a ViT model pretrained on ImageNet-1K using WASI on the Pets dataset. We use six values of $\varepsilon\in\{0.4, 0.5, 0.6, 0.7, 0.8, 0.9\}$ and run each configuration with five different random seeds $\{42, 100, 233, 234, 235\}$ (randomly chosen).
>
> The table below summarizes the results across five seeds, reported as mean $\pm$ standard deviation:
> |**$\varepsilon$**|**0.4**|**0.5**|**0.6**|**0.7**|**0.8**|**0.9**|
> |---|:---:|:---:|:---:|:---:|:---:|:---:|
> |**Acc (%)**|19.44 ± 1.10|21.76 ± 1.04|27.15 ± 1.01|33.26 ± 0.84|59.96 ± 1.03|85.50 ± 0.68|
> |**Training Mem (MB)**|28.60 ± 0.00|36.24 ± 0.00|49.48 ± 0.00|66.36 ± 0.00|89.31 ± 0.00|125.95 ± 0.00|
> |**Inference Mem (MB)**|25.62 ± 0.00|33.25 ± 0.00|46.49 ± 0.00|63.37 ± 0.00|86.32 ± 0.00|122.96 ± 0.00|
> |**Training FLOPs**|2.47E+11 ± 0.00|3.26E+11 ± 0.00|4.24E+11 ± 0.00|5.50E+11 ± 0.00|7.21E+11 ± 0.00|9.96E+11 ± 0.00|
> |**Inference FLOPs**|1.25E+11 ± 0.00|1.68E+11 ± 0.00|2.22E+11 ± 0.00|2.91E+11 ± 0.00|3.85E+11 ± 0.00|5.38E+11 ± 0.00|
> - The accuracy shows only minor variance across different seeds.
> - We emphasize that **WASI is fundamentally a combination of SVD and Gram–Schmidt orthogonalization** (Algorithms 1 and 2 in our paper), both of which are deterministic procedures. In theory, this means **WASI itself does not depend on random seeds**. That explains why the resulting resource consumption is identical across seeds.
> - The small variance in accuracy mainly comes from the stochastic components of the training pipeline (e.g., data shuffling, dropout, layer-norm noise, and optimizer randomness), rather than from the WASI algorithm.
>
> We will add this table to the Appendix. You can reproduce this experiment by modifying the ``--seed_everything`` variable in the corresponding bash script inside the `main/scripts` folder of the supplementary code.

---

> > ### Comment · Reviewer_ZTZ2 · 2025-11-21
> >
> > Thanks for addressing my concerns. I think the work is decent and mainly incremental. So I would stick to my original rating.

---

### Author Response · Authors · 2025-12-01
**Summary of the rebuttal phase**

Dear Area Chair,

Thank you for your effort in coordinating the review process. Because the rebuttal phase was interrupted, we provide below a concise summary of the discussion for your convenience.

Importantly, we **fully addressed all concerns** raised by the reviewers. Therefore, the overall score distribution was raised from **(6, 6, 6) to (6, 8, 8)**.

For completeness and transparency, we briefly summarize each reviewer’s concerns and our responses:

---

**[Reviewer ZTZ2 – Score: 6 → 6]**

* **Novelty of the method.** We clarified our contributions and argued that our approach is the first to exploit subspace stability to design an efficient on-device training algorithm.
* **Major gains are for training memory.** We emphasized that reducing training memory is the primary goal of our method, as stated in the title and throughout the paper.
* **No power consumption report.** We added new experiments with consistent results.
* **Missing results across seeds.** We pointed out that seed-dependent results are already reported in Figure 9 (Appendix B.2) and additionally provided a numerical table over 5 random seeds in the rebuttal.

---

**[Reviewer qswV – Score: 6 → 8]**

* **Request for more experiments on more edge devices.** We added further experiments on 3 more edge devices, showing consistent latency reductions across devices.
* **No table for final accuracy.** We explained our choice of Pareto-style plots for readability and fairness with prior work, and we provided an additional table as requested.
* **Attention-layer coverage only in the appendix.** We clarified that the main text focuses on MLP blocks to ensure fair comparison with baselines, which was explicitly stated in the paper.
* **Concerns about federated-learning applications.** We clarified that our work is explicitly targeted at single-device on-device fine-tuning.
* **Request for micro-benchmark of one WASI iteration.** We provided additional experiments with detailed explanation.
* **Sensitivity of the results.** We provided detailed explanation.
* **Request for strategy to define memory or latency constraints for the method.** We provided a step-by-step strategy.

---

**[Reviewer cFHJ – Score: 6 → 8]**

* **Novelty of the method.** We clarified our contributions and argued that our method is the first of its kind. The Reviewer claimed that our idea have been explored without providing a concrete reference.
* **Instability of the method.** We explained that across all our datasets, the trade-off between accuracy and resource usage is stable and monotonic with respect to $\varepsilon$. We also asked the Reviewer for specific examples of instability, but none were provided.
* **Experiments on relatively small datasets.** We clarified that our setting follows the standard on-device learning scenario, where a large pretrained model is fine-tuned on smaller, task-specific datasets on the edge device.
* **Request for training from scratch on large-scale datasets.** We provided additional experimental results.
* **Request for strategy to define $\varepsilon$.** We provided detailed explanation with step-by-step strategy to choose $\varepsilon$.

---

All additional results will be included in the revised version of the paper, and we will release all related code upon the paper acceptance.

We hope this summary helps you to contextualize the reviews, the rebuttal, and the subsequent score changes.

Sincerely,

The Authors

---

### Meta-Review · Area_Chair_qhTH · 2026-01-14

**Summary:**

This paper proposes WASI (Weight–Activation Subspace Iteration), a subspace-based training framework for Transformer models that enables efficient on-device fine-tuning by restricting both weight and activation updates to a learned low-rank subspace. By combining Weight Subspace Iteration with Activation Subspace Iteration, WASI substantially alleviates backpropagation memory bottlenecks and reduces computational cost while preserving model accuracy. Experiments show that WASI achieves up to noticeable memory and FLOPs reductions.

**Reviewer Concerns:**

Overall, the reviewers acknowledge the technical soundness of the approach but raise consistent concerns about limited novelty, noting that the core ideas—low-dimensional subspace restriction and low-rank approximations—have been explored in prior work, making the contribution largely incremental and engineering-oriented. The experimental validation is considered insufficient, with limited edge-device diversity (primarily Raspberry Pi 5), small-scale datasets, missing variance analysis across random seeds, and a lack of power/energy consumption measurements, all of which weaken claims about robustness and real-world edge applicability. Additionally, comparative evaluations and reporting clarity need improvement, including clearer presentation of final accuracy, broader comparisons (e.g., attention layers, heterogeneous/non-IID settings), sensitivity to hyperparameters, and more informative visualizations.

**Reviewer Scores:**

During the rebuttal period, key concerns, particularly regarding technical novelty, were adequately addressed and accepted by the reviewers, leading most of them to raise their scores. Consequently, the overall evaluation trend shifted toward a more positive recommendation.

---

### Decision · Program_Chairs · 2026-01-26

Accept (Oral)